# Uncertainty-Aware Meta-Learning for Multimodal Task Distributions

## Abstract

Meta-learning or *learning to learn* is a popular approach for learning new tasks with limited data (i.e., *few-shot learning*) by leveraging the commonalities among different tasks. However, meta-learned models can perform poorly when context data is limited, or when data is drawn from an out-of-distribution (OoD) task. Especially in safety-critical settings, this necessitates an uncertainty-aware approach to meta-learning. In addition, the often multimodal nature of task distributions can pose unique challenges to meta-learning methods. In this work, we present UNLIMITD (*uncertainty-aware meta-learning for multimodal task distributions*), a novel method for meta-learning that (1) makes probabilistic predictions on in-distribution tasks efficiently, (2) is capable of detecting OoD context data at test time, and (3) performs on heterogeneous, multimodal task distributions. To achieve this goal, we take a probabilistic perspective and train a parametric, tuneable distribution over tasks on the meta-dataset. We construct this distribution by performing Bayesian inference on a linearized neural network, leveraging Gaussian process theory. We demonstrate that UNLIMITD's predictions compare favorably to, and outperform in most cases, the standard baselines, especially in the low-data regime. Furthermore, we show that UNLIMITD is effective in detecting data from OoD tasks. Finally, we confirm that both of these findings continue to hold in the multimodal task-distribution setting.

## 1 Introduction

Learning to learn is essential in human intelligence but is still a wide area of research in machine learning. *Meta-learning* has emerged as a popular approach to enable models to perform well on new tasks using limited data. It involves first a *meta-training* process, when the model learns valuable features from a set of tasks. Then, at test time, using only few datapoints from a new, unseen task, the model (1) *adapts* to this new task (i.e., performs *few-shot learning* with *context data*), and then (2) *infers* by making predictions on new, unseen *query inputs* from the same task. A popular baseline for meta-learning, which has attracted a large amount of attention, is Model-Agnostic Meta-Learning (MAML) (Finn et al., 2017), in which the adaptation process consists of fine-tuning the parameters of the model via gradient descent.

However, meta-learning methods can often struggle in several ways when deployed in challenging real-world scenarios. First, when context data is too limited to fully identify the test-time task, accurate prediction can be challenging. As these predictions can be untrustworthy, this necessitates the development of meta-learning methods that can express uncertainty during adaptation (Yoon et al., 2018; Harrison et al., 2018). In addition, meta-learning models may not successfully adapt to "unusual" tasks, i.e., when test-time context data is drawn from an *out-of-distribution* (OoD) task not well represented in the training dataset (Jeong & Kim, 2020; Iwata & Kumagai, 2022). Finally, special care has to be taken when learning tasks that have a large degree of heterogeneity. An important example is the case of tasks with a *multimodal* distribution, i.e., when there are no common features shared across all the tasks, but the tasks can be broken down into subsets (modes) in a way that the ones from the same subset share common features (Vuorio et al., 2019).

**Our contributions.** We present UNLIMITD (*uncertainty-aware meta-learning for multimodal task distributions*), a novel meta-learning method that leverages probabilistic tools to address the aforementioned issues. Specifically, UNLIMITD models the true distribution of tasks with a learnable distribution constructed over a linearized neural network and uses analytic Bayesian inference to perform uncertainty-aware adaption. We present three variants (namely, UNLIMITD-I, UNLIM-

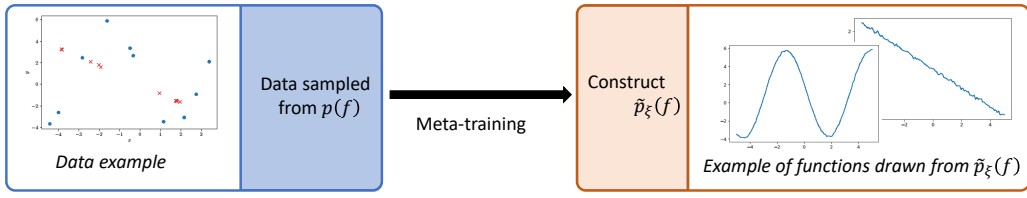

Figure 1: The true task distribution $p(f)$ can be multimodal, i.e., containing multiple clusters of tasks (e.g., lines and sines). Our approach UNLIMITD fits $p(f)$ with a parametric, tuneable distribution $\tilde{p}_\xi(f)$ yielded by Bayesian linear regression on a linearized neural network.

ITD-R, and UNLIMITD-F) that reflect a trade-off between learning a rich prior distribution over the weights and maintaining the full expressivity of the network; we show that UNLIMITD-F strikes a balance between the two, making it the most appealing variant. Finally, we demonstrate that (1) our method allows for efficient probabilistic predictions on in-distribution tasks, that compare favorably to, and in most cases outperform, the existing baselines, (2) it is effective in detecting context data from OoD tasks at test time, and that (3) both these findings continue to hold in the multimodal task-distribution setting.

The rest of the paper is organized as follows. Section 2 formalizes the problem. Section 3 presents background information on the linearization of neural networks and Bayesian linear regression. We detail our approach and its three variants in Section 4. We discuss related work in detail in Section 5. Finally, we present our experimental results concerning the performance of UNLIMITD in Section 6 and conclude in Section 7.

## 2 PROBLEM STATEMENT

A task $\mathcal{T}^i$ consists of a function $f_i$ from which data is drawn. At test time, the prediction steps are broken down into (1) *adaptation*, that is identifying $f_i$ using $K$ context datapoints $(\boldsymbol{X}^i, \boldsymbol{Y}^i)$ from the task, and (2) *inference*, that is making predictions for $f_i$ on the *query inputs* $\boldsymbol{X}_*^i$. Later the predictions can be compared with the *query ground-truths* $\boldsymbol{Y}_*^i$ to estimate the quality of the prediction, for example in terms of mean squared error (MSE). The meta-training consists in learning valuable features from a *cluster of tasks*, which is a set of similar tasks (e.g., sines with different phases and amplitudes but same frequency), so that at test time the predictions can be accurate on tasks from the same cluster. We take a probabilistic, functional perspective and represent a cluster by $p(f)$, a theoretical distribution over the function space that describes the probability of a task belonging to the cluster. Learning $p(f)$ is appealing, as it allows for performing OoD detection in addition to making predictions. Adaptation amounts to computing the conditional distribution given test context data, and one can obtain an uncertainty metric by evaluating the negative log-likelihood (NLL) of the context data under $p(f)$.

Thus, our goal is to construct a parametric, learnable functional distribution $\tilde{p}_\xi(f)$ that approaches the theoretical distribution $p(f)$, with a structure that allows tractable conditioning and likelihood computation, even in deep learning contexts. In practice, however, we are not given $p(f)$, but only a meta-training dataset $\mathcal{D}$ that we assume is sampled from $p(f)$: $\mathcal{D} = \{(\widetilde{\boldsymbol{X}}^i, \widetilde{\boldsymbol{Y}}^i)\}_{i=1}^N$, where $N$ is the number of tasks available during training, and $(\widetilde{\boldsymbol{X}}^i, \widetilde{\boldsymbol{Y}}^i) \sim \mathcal{T}^i$ is the entire pool of data from which we can draw subsets of context data $(\boldsymbol{X}^i, \boldsymbol{Y}^i)$. Consequently, in the meta-training phase, we aim to optimize $\tilde{p}_\xi(f)$ to capture properties of $p(f)$, using only the samples in $\mathcal{D}$.

Once we have $\tilde{p}_\xi(f)$, we can evaluate it both in terms of how it performs for few-shot learning (by comparing the predictions with the ground truths in terms of MSE), as well as for OoD detection (by measuring how well the NLL of context data serves to classify in-distribution tasks against OoD tasks, measured via the AUC-ROC score).

## 3 BACKGROUND

### 3.1 BAYESIAN LINEAR REGRESSION AND GAUSSIAN PROCESSES

Efficient Bayesian meta-learning requires a tractable inference process at test time. In general, this is only possible analytically in a few cases. One of them is the Bayesian linear regression with Gaussian noise and a Gaussian prior on the weights. Viewing it from a nonparametric, functional approach, this model is equivalent to a Gaussian process (GP) (Rasmussen & Williams, 2005).

Let $\boldsymbol{X} = (\boldsymbol{x}_1, \ldots, \boldsymbol{x}_K) \in \mathbb{R}^{N_x \times K}$ be a batch of $K$ $N_x$-dimensional inputs, and let $\boldsymbol{y} = (\boldsymbol{y}_1, \ldots, \boldsymbol{y}_K) \in \mathbb{R}^{N_y K}$ be a vectorized batch of $N_y$-dimensional outputs. In the Bayesian linear regression model, these quantities are related according to $\boldsymbol{y} = \phi(\boldsymbol{X})^\top \hat{\boldsymbol{\theta}} + \varepsilon \in \mathbb{R}^{N_y K}$ where $\hat{\boldsymbol{\theta}} \in \mathbb{R}^P$ are the weights of the model, and the inputs are mapped via $\phi : \mathbb{R}^{N_x \times K} \to \mathbb{R}^{P \times N_y K}$. Notice how this is a generalization of the usual one-dimensional linear regression ($N_y = 1$).

If we assume a Gaussian prior on the weights $\hat{\boldsymbol{\theta}} \sim \mathcal{N}(\boldsymbol{\mu}, \boldsymbol{\Sigma})$ and a Gaussian noise $\varepsilon \sim \mathcal{N}(\boldsymbol{0}, \boldsymbol{\Sigma}_\varepsilon)$ with $\boldsymbol{\Sigma}_\varepsilon = \sigma_\varepsilon^2 \boldsymbol{I}$, then the model describes a multivariate Gaussian distribution on $\boldsymbol{y}$ for any $\boldsymbol{X}$. Equivalently, this means that this model describes a GP distribution over functions, with mean and covariance function (or kernel)

$$\boldsymbol{\mu}_{\text{prior}}(\boldsymbol{x}_t) = \phi(\boldsymbol{x}_t)^\top \boldsymbol{\mu},$$
$$\text{cov}_{\text{prior}}(\boldsymbol{x}_{t_1}, \boldsymbol{x}_{t_2}) = \phi(\boldsymbol{x}_{t_1})^\top \boldsymbol{\Sigma} \phi(\boldsymbol{x}_{t_2}) + \boldsymbol{\Sigma}_\varepsilon := k_{\boldsymbol{\Sigma}}(\boldsymbol{x}_{t_1}, \boldsymbol{x}_{t_2}) + \boldsymbol{\Sigma}_\varepsilon. \tag{1}$$

This GP enables tractable computation of the likelihood of any batch of data $(\boldsymbol{X}, \boldsymbol{Y})$ given this distribution over functions. The structure of this distribution is governed by the feature map $\phi$ and the prior over the weights, specified by $\boldsymbol{\mu}$ and $\boldsymbol{\Sigma}$.

This distribution can also easily be conditioned to perform inference. Given a batch of data, the posterior predictive distribution is also a GP, with an updated mean and covariance function

$$\boldsymbol{\mu}_{\text{post}}(\boldsymbol{x}_{t*}) = k_{\boldsymbol{\Sigma}}(\boldsymbol{x}_{t*}, \boldsymbol{X}) \left( k_{\boldsymbol{\Sigma}}(\boldsymbol{X}, \boldsymbol{X}) + \boldsymbol{\Sigma}_\varepsilon \right)^{-1} \boldsymbol{Y},$$
$$\text{cov}_{\text{post}}(\boldsymbol{x}_{t_1*}, \boldsymbol{x}_{t_2*}) = k_{\boldsymbol{\Sigma}}(\boldsymbol{x}_{t_1*}, \boldsymbol{x}_{t_2*}) - k_{\boldsymbol{\Sigma}}(\boldsymbol{x}_{t_1*}, \boldsymbol{X}) \left( k_{\boldsymbol{\Sigma}}(\boldsymbol{X}, \boldsymbol{X}) + \boldsymbol{\Sigma}_\varepsilon \right)^{-1} k_{\boldsymbol{\Sigma}}(\boldsymbol{X}, \boldsymbol{x}_{t_2*}). \tag{2}$$

Here, $\boldsymbol{\mu}_{\text{post}}(\boldsymbol{X}_*)$ represents our model's adapted predictions for the test data, which we can compare to $\boldsymbol{Y}_*$ to evaluate the quality of our predictions, for example, via mean squared error (assuming that test data is clean, following Rasmussen & Williams (2005)). The diagonal of $\text{cov}_{\text{post}}(\boldsymbol{X}_*, \boldsymbol{X}_*)$ can be interpreted as a per-input level of confidence that captures the ambiguity in making predictions with only a limited amount of context data.

### 3.2 THE LINEARIZATION OF A NEURAL NETWORK YIELDS AN EXPRESSIVE LINEAR REGRESSION MODEL

As discussed, the choice of feature map $\phi$ plays an important role in specifying a linear regression model. In the deep learning context, recent work has demonstrated that the linear model obtained when linearizing a deep neural network with respect to its weights at initialization, wherein the Jacobian of the network operates as the feature map, can well approximate the training behavior of wide nonlinear deep neural networks (Jacot et al., 2018; Neal, 1996; Azizan et al., 2021; Lee et al., 2017).

Let $f$ be a neural network $f : (\boldsymbol{\theta}, \boldsymbol{x}_t) \mapsto \boldsymbol{y}_t$, where $\boldsymbol{\theta} \in \mathbb{R}^P$ are the parameters of the model, $\boldsymbol{x} \in \mathbb{R}^{N_x}$ is an input and $\boldsymbol{y} \in \mathbb{R}^{N_y}$ an output. The linearized network (w.r.t. the parameters) on $\boldsymbol{\theta}_0$ is

$$f(\boldsymbol{\theta}, \boldsymbol{x}_t) - f(\boldsymbol{\theta}_0, \boldsymbol{x}_t) \approx \boldsymbol{J}_{\boldsymbol{\theta}}(f)(\boldsymbol{\theta}_0, \boldsymbol{x}_t)(\boldsymbol{\theta} - \boldsymbol{\theta}_0),$$

where $\boldsymbol{J}_{\boldsymbol{\theta}}(f)(\cdot, \cdot) : \mathbb{R}^P \times \mathbb{R}^{N_x} \to \mathbb{R}^{N_y \times P}$ is the Jacobian of the network (w.r.t. the parameters).

In the case where the model accepts a batch of $K$ inputs $\boldsymbol{X} = (\boldsymbol{x}_1, \ldots, \boldsymbol{x}_K)$ and returns $\boldsymbol{Y} = (\boldsymbol{y}_1, \ldots, \boldsymbol{y}_K)$, we generalize $f$ into $g : \mathbb{R}^P \times \mathbb{R}^{N_x \times K} \to \mathbb{R}^{N_y \times K}$, with $\boldsymbol{Y} = g(\boldsymbol{\theta}, \boldsymbol{X})$. Consequently, we generalize the linearization:

$$g(\boldsymbol{\theta}, \boldsymbol{X}) - g(\boldsymbol{\theta}_0, \boldsymbol{X}) \approx \boldsymbol{J}(\boldsymbol{\theta}_0, \boldsymbol{X})(\boldsymbol{\theta} - \boldsymbol{\theta}_0),$$

where $\boldsymbol{J}(\cdot, \cdot) : \mathbb{R}^P \times \mathbb{R}^{N_x \times K} \to \mathbb{R}^{N_y K \times P}$ is a shorthand for $\boldsymbol{J}_{\boldsymbol{\theta}}(g)(\cdot, \cdot)$. Note that we have implicitly vectorized the outputs, and throughout the work, we will freely interchange the matrices $\mathbb{R}^{N_y \times K}$ and the vectorized matrices $\mathbb{R}^{N_y K}$.

This linearization can be viewed as the $N_y K$-dimensional linear regression

$$\boldsymbol{z} = \phi_{\boldsymbol{\theta}_0}(\boldsymbol{X})^\top \hat{\boldsymbol{\theta}} \in \mathbb{R}^{N_y K}, \tag{3}$$

where the feature map $\phi_{\boldsymbol{\theta}_0}(\cdot) : \mathbb{R}^{N_x \times K} \rightarrow \mathbb{R}^{P \times N_y K}$ is the transposed Jacobian $\boldsymbol{J}(\boldsymbol{\theta}_0, \cdot)^\top$. The parameters of this linear regression $\hat{\boldsymbol{\theta}} = (\boldsymbol{\theta} - \boldsymbol{\theta}_0)$ are the *correction* to the parameters chosen as the linearization point. Equivalently, this can be seen as a kernel regression with the kernel $k_{\boldsymbol{\theta}_0}(\boldsymbol{X}_1, \boldsymbol{X}_2) = \boldsymbol{J}(\boldsymbol{\theta}_0, \boldsymbol{X}_1)\boldsymbol{J}(\boldsymbol{\theta}_0, \boldsymbol{X}_2)^\top$, which is commonly referred to as the Neural Tangent Kernel (NTK) of the network. Note that the NTK depends on the linearization point $\boldsymbol{\theta}_0$. Building on these ideas, Maddox et al. (2021) show that the NTK obtained via linearizing a DNN *after* it has been trained on a task yields a GP that is well-suited for adaptation and fine-tuning to new, similar tasks. Furthermore, they show that networks trained on similar tasks tend to have similar Jacobians, suggesting that neural network linearization can yield an effective model for multi-task contexts such as meta-learning. In this work, we leverage these insights to construct our parametric functional distribution $\tilde{p}_\xi(f)$ via linearizing a neural network model.

# 4   OUR APPROACH: UNLIMITD

In this section, we describe our meta-learning algorithm UNLIMITDand the construction of a parametric functional distribution $\tilde{p}_\xi(f)$ that can model the true underlying distribution over tasks $p(f)$. First, we focus on the single cluster case, where a Gaussian process structure on $\tilde{p}_\xi(f)$ can effectively model the true distribution of tasks, and detail how we can leverage meta-training data $\mathcal{D}$ from a single cluster of tasks to train the parameters $\xi$ of our model. Next, we will generalize our approach to the multimodal setting, with more than one cluster of tasks. Here, we construct $\tilde{p}_\xi(f)$ as a mixture of GPs and develop a training approach that can automatically identify the clusters present in the training dataset without requiring the meta-training dataset to contain any additional structure such as cluster labels.

## 4.1   TRACTABLY STRUCTURING THE PRIOR PREDICTIVE DISTRIBUTION OVER FUNCTIONS VIA A GAUSSIAN DISTRIBUTION OVER THE WEIGHTS

In our approach, we choose $\tilde{p}_\xi(f)$ to be the GP distribution over functions that arises from a Gaussian prior on the weights of the linearization of a neural network (equation 3). Consider a particular task $\mathcal{T}^i$ and a batch of $K$ context data $(\boldsymbol{X}^i, \boldsymbol{Y}^i)$. The resulting prior predictive distribution, derived from equation 1 after evaluating on the context inputs, is $\boldsymbol{Y}|\boldsymbol{X}^i \sim \mathcal{N}(\boldsymbol{\mu}_{\boldsymbol{Y}|\boldsymbol{X}^i}, \boldsymbol{\Sigma}_{\boldsymbol{Y}|\boldsymbol{X}^i})$, where

$$\boldsymbol{\mu}_{\boldsymbol{Y}|\boldsymbol{X}^i} = \boldsymbol{J}(\boldsymbol{\theta}_0, \boldsymbol{X}^i)\boldsymbol{\mu}, \quad \boldsymbol{\Sigma}_{\boldsymbol{Y}|\boldsymbol{X}^i} = \boldsymbol{J}(\boldsymbol{\theta}_0, \boldsymbol{X}^i)\boldsymbol{\Sigma}\boldsymbol{J}(\boldsymbol{\theta}_0, \boldsymbol{X}^i)^\top + \boldsymbol{\Sigma}_\varepsilon). \tag{4}$$

In this setup, the parameters $\xi$ of $\tilde{p}_\xi(f)$ that we wish to optimize are the linearization point $\boldsymbol{\theta}_0$, and the parameters of the prior over the weights $(\boldsymbol{\mu}, \boldsymbol{\Sigma})$. Given this Gaussian prior, it is straightforward to compute the joint NLL of the context labels $\boldsymbol{Y}^i$,

$$\text{NLL}(\boldsymbol{X}^i, \boldsymbol{Y}^i) = \frac{1}{2}\left( \left\| \boldsymbol{Y}^i - \boldsymbol{\mu}_{\boldsymbol{Y}|\boldsymbol{X}^i} \right\|_{\boldsymbol{\Sigma}_{\boldsymbol{Y}|\boldsymbol{X}^i}^{-1}}^2 + \log\det\boldsymbol{\Sigma}_{\boldsymbol{Y}|\boldsymbol{X}^i} + N_y K \log 2\pi \right). \tag{5}$$

The NLL (a) serves as a loss function quantifying the quality of $\xi$ during training and (b) serves as an uncertainty signal at test time to evaluate whether context data $(\boldsymbol{X}^i, \boldsymbol{Y}^i)$ is OoD. Given this model, *adaptation* is tractable as we can condition this GP on the context data analytically. In addition, we can efficiently make probabilistic predictions by evaluating the mean and covariance of the resulting posterior predictive distribution on the query inputs, using equation 2.

### 4.1.1   PARAMETERIZING THE PRIOR COVARIANCE OVER THE WEIGHTS

When working with deep neural networks, the number of weights $P$ can surpass $10^6$. While it remains tractable to deal with $\boldsymbol{\theta}_0$ and $\boldsymbol{\mu}$, whose memory footprint grows linearly with $P$, it can quickly become intractable to make computations with (let alone store) a dense prior covariance matrix over the weights $\boldsymbol{\Sigma} \in \mathbb{R}^{P \times P}$. Thus, we must impose some structural assumptions on the prior covariance to scale to deep neural network models.

**Imposing a unit covariance.** One simple way to tackle this issue would be to remove $\boldsymbol{\Sigma}$ from the learnable parameters $\xi$, i.e. fixing it to be the identity $\boldsymbol{\Sigma} = \boldsymbol{I}_P$. In this case, $\xi = (\boldsymbol{\theta}_0, \boldsymbol{\mu})$. This

computational benefit comes at the cost of model expressivity, as we lose a degree of freedom in how we can optimize our learned prior distribution $\tilde{p}_\xi(f)$. In particular, we are unable to choose a prior over the weights of our model that captures correlations between elements of the feature map.

**Learning a low-dimensional representation of the covariance.** An alternative is to learn a low-rank representation of $\Sigma$, allowing for a learnable weight-space prior covariance that can encode correlations. Specifically, we consider a covariance of the form $\Sigma = \boldsymbol{Q}^\top \text{diag}(\boldsymbol{s}^2)\boldsymbol{Q}$, where $\boldsymbol{Q}$ is a fixed projection matrix on an $s$-dimensional subspace of $\mathbb{R}^P$, while $\boldsymbol{s}^2$ is learnable. In this case, the parameters that are learned are $\xi = (\boldsymbol{\theta}_0, \boldsymbol{\mu}, \boldsymbol{s})$. We define $\boldsymbol{S} := \text{diag}(\boldsymbol{s}^2)$. The computation of the covariance of the prior predictive (equation 4) could then be broken down into two steps:

$$\begin{cases} A := \boldsymbol{J}(\boldsymbol{\theta}_0, \boldsymbol{X}^i)\boldsymbol{Q}^\top \\ \boldsymbol{J}(\boldsymbol{\theta}_0, \boldsymbol{X}^i)\Sigma\boldsymbol{J}(\boldsymbol{\theta}_0, \boldsymbol{X}^i)^\top = A\boldsymbol{S}A^\top \end{cases}$$

which requires a memory footprint of $O(P(s + N_y K))$, if we include the storage of the Jacobian. Because $N_y K \ll P$ in typical deep learning contexts, it suffices that $s \ll P$ so that it becomes tractable to deal with this new representation of the covariance.

**A trade-off between feature-map expressiveness and learning a rich prior over the weights.** Note that even if a low-dimensional representation of $\Sigma$ enriches the prior distribution over the weights, it also restrains the expressiveness of the feature map in the kernel by projecting the $P$-dimensional features $\boldsymbol{J}(\boldsymbol{\theta}_0, \boldsymbol{X})$ on a subspace of size $s \ll P$ via $\boldsymbol{Q}$. This presents a trade-off: we can use the full feature map, but limit the weight-space prior covariance to be the identity matrix by keeping $\Sigma = \boldsymbol{I}$ (case UNLIMITD-I). Alternatively, we could learn a low-rank representation of $\Sigma$ by randomly choosing $s$ orthogonal directions in $\mathbb{R}^P$, with the risk that they could limit the expressiveness of the feature map if the directions are not relevant to the problem that is considered (case UNLIMITD-R). As a compromise between these two cases, we can choose the projection matrix more intelligently and project to the most impactful subspace of the full feature map — in this way, we can reap the benefits of a tuneable prior covariance while minimizing the useful features that the projection drops. To select this subspace, we construct this projection map by choosing the top $s$ eigenvectors of the Fisher information matrix (FIM) evaluated on the training dataset $\mathcal{D}$ (case UNLIMITD-F). Recent work has shown that the FIM for deep neural networks tends to have rapid spectral decay (Sharma et al., 2021), which suggests that keeping only a few of the top eigenvectors of the FIM is enough to encode an expressive task-tailored prior. See Appendix A.1 for more details.

### 4.1.2 GENERALIZING THE STRUCTURE TO A MIXTURE OF GAUSSIANS

When learning on multiple clusters of tasks, $p(f)$ can become non-unimodal, and thus cannot be accurately described by a single GP. Instead, we can capture this multimodality by structuring $\tilde{p}_\xi(f)$ as a *mixture* of Gaussian processes.

**Building a more general structure.** We assume that at train time, a task $\mathcal{T}^i$ comes from any cluster $\{\mathcal{C}_j\}_{j=1}^{j=\alpha}$ with equal probability. Thus, we choose to construct $\tilde{p}_\xi(f)$ as an equal-weighted mixture of $\alpha$ Gaussian processes.

For each element of the mixture, the structure is similar to the single cluster case, where the parameters of the cluster's weight-space prior are given by $(\boldsymbol{\mu}_j, \Sigma_j)$. We choose to have both the projection matrix $\boldsymbol{Q}$ and the linearization point $\boldsymbol{\theta}_0$ (and hence, the feature map $\phi(\cdot) = \boldsymbol{J}(\boldsymbol{\theta}_0, \cdot)$) shared across the clusters. This yields improved computational efficiency, as we can compute the projected features once, simultaneously, for all clusters. This yields the parameters $\xi_\alpha = (\boldsymbol{\theta}_0, \boldsymbol{Q}, (\boldsymbol{\mu}_1, \boldsymbol{s}_1), \ldots, (\boldsymbol{\mu}_\alpha, \boldsymbol{s}_\alpha))$.

This can be viewed as a mixture of linear regression models, with a common feature map but separate, independent prior distributions over the weights for each cluster. These separate distributions are encoded using the low-dimensional representations $\boldsymbol{S}_j$ for each $\Sigma_j$. Notice how this is a generalization of the single cluster case, for when $\alpha = 1$, $\tilde{p}_\xi(f)$ becomes a Gaussian and $\xi_\alpha = \xi^1$.

**Prediction and likelihood computation.** The NLL of a batch of inputs under this mixture model can be computed as

$$\text{NLL}_{\text{mixt}}(\boldsymbol{X}^i, \boldsymbol{Y}^i) = \log \alpha - \log \text{sum} \exp(-\text{NLL}_1(\boldsymbol{X}^i, \boldsymbol{Y}^i), \ldots, -\text{NLL}_\alpha(\boldsymbol{X}^i, \boldsymbol{Y}^i)), \quad (6)$$

---

[1] In theory, it is possible to drop $\boldsymbol{Q}$ and extend the identity covariance case to the multi-cluster setting, however, this leads to each cluster having an identical covariance function, and thus is not effective at modeling heterogeneous behaviors among clusters.

---

**Algorithm 1** UNLIMITD-I: meta-training with identity prior covariance

---

1: Initialize $\boldsymbol{\theta}_0, \boldsymbol{\mu}$.
2: **for all** epoch **do**
3:     Sample $n$ tasks $\{\mathcal{T}^i, (\boldsymbol{X}^i, \boldsymbol{Y}^i)\}_{i=1}^{i=n}$
4:     **for all** $\mathcal{T}^i, (\boldsymbol{X}^i, \boldsymbol{Y}^i)$ **do**
5:         $NLL_i \leftarrow$ GAUSSNLL$(\boldsymbol{Y}^i; \boldsymbol{J\mu}, \boldsymbol{JJ}^\top + \boldsymbol{\Sigma}_\varepsilon)$                 $\triangleright \boldsymbol{J} = \boldsymbol{J}(\boldsymbol{\theta}_0, \boldsymbol{X}^i)$
6:     **end for**
7:     Update $\boldsymbol{\theta}_0, \boldsymbol{\mu}$ with $\nabla_{\boldsymbol{\theta}_0 \cup \boldsymbol{\mu}} \sum_i NLL_i$
8: **end for**

---

**Algorithm 2** UNLIMITD-R and UNLIMITD-F: meta-training with a learnt covariance

---

1: **if** using random projections **then**
2:     Find random projection $\boldsymbol{Q}$
3:     Initialize $\boldsymbol{\theta}_0, \boldsymbol{\mu}, \boldsymbol{s}$
4: **else if** using FIM-based projections **then**
5:     Find intermediate $\boldsymbol{\theta}_0, \boldsymbol{\mu}$ with UNLIMITD-I                $\triangleright$ see Alg. 1
6:     Find $\boldsymbol{Q}$ via FIMPROJ($\boldsymbol{s}$); initialize $\boldsymbol{s}$.               $\triangleright$ see Alg. 3
7: **end if**
8: **for all** epoch **do**
9:     Sample $n$ tasks $\{\mathcal{T}^i, (\boldsymbol{X}^i, \boldsymbol{Y}^i)\}_{i=1}^{i=n}$
10:     **for all** $\mathcal{T}^i, (\boldsymbol{X}^i, \boldsymbol{Y}^i)$ **do**
11:         $NLL_i \leftarrow$ GAUSSNLL$(\boldsymbol{Y}^i; \boldsymbol{J\mu}, \boldsymbol{JQ}^\top \text{diag}(\boldsymbol{s}^2) \boldsymbol{QJ}^\top + \boldsymbol{\Sigma}_\varepsilon)$     $\triangleright \boldsymbol{J} = \boldsymbol{J}(\boldsymbol{\theta}_0, \boldsymbol{X}^i)$
12:     **end for**
13:     Update $\boldsymbol{\theta}_0, \boldsymbol{\mu}, \boldsymbol{s}$ with $\nabla_{\boldsymbol{\theta}_0 \cup \boldsymbol{\mu} \cup \boldsymbol{s}} \sum_i NLL_i$
14: **end for**

---

where $\text{NLL}_j(\boldsymbol{X}^i, \boldsymbol{Y}^i)$ is the NLL with respect to each individual Gaussian, as computed in equation 5, and $\log \text{sum} \exp$ computes the logarithm of the sum of the exponential of the arguments, taking care to avoid underflow issues.

To make exact predictions, we would require conditioning this mixture model. As this is not directly tractable, we propose to first *infer the cluster* from which a task comes from, by identifying the Gaussian $\mathcal{G}_{j_0}$ that yields the highest likelihood for the context data $(\boldsymbol{X}^i, \boldsymbol{Y}^i)$. Then, we can *adapt* by conditioning $\mathcal{G}_{j_0}$ with the context data and finally *infer* by evaluating the resulting posterior distribution on the queried inputs $\boldsymbol{X}^i_*$.

### 4.2 META-TRAINING THE PARAMETRIC TASK DISTRIBUTION

The key to our meta-learning approach is to estimate the quality of $\tilde{p}_\xi(f)$ via the NLL of context data from training tasks, and use its gradients to update the parameters of the distribution $\xi$. Optimizing this loss over tasks in the dataset draws $\tilde{p}_\xi(f)$ closer to the empirical distribution present in the dataset, and hence towards the true distribution $p(f)$.

We present three versions of UNLIMITD, depending on the choice of structure of the prior covariance over the weights (see Section 4.1.1 for more details). UNLIMITD-I (Algorithm 1) is the meta-training with the fixed identity prior covariance. UNLIMITD-R and UNLIMITD-F (Algorithm 2) learn a low-dimensional representation of that prior covariance, either with random projections or with FIM-based projections.

**Computing the likelihood.** In the algorithms, the function GAUSSNLL$(\boldsymbol{Y}^i; m, K)$ stands for NLL of $\boldsymbol{Y}^i$ under the Gaussian $\mathcal{N}(m, K)$ (see equation 5). In the mixture case, we instead use MIXTNLL, which wraps equation 6 and calls GAUSSNLL for the individual NLL computations (see discussion in Section 4.1.2). In this case, $\boldsymbol{\mu}$ becomes $\{\boldsymbol{\mu}_j\}_{j=1}^{j=\alpha}$ and $\boldsymbol{s}$ becomes $\{\boldsymbol{s}_j\}_{j=1}^{j=\alpha}$ when applicable.

**Finding the FIM-based projections.** The FIM-based projection matrix aims to identify the elements of $\phi = \boldsymbol{J}(\boldsymbol{\theta}_0, \boldsymbol{X})$ that are most relevant for the problem (see Section 4.1.1 and Appendix A.1). However, this feature map evolves during training, because it is $\boldsymbol{\theta}_0$-dependent. How do we ensure that the directions we choose for $\boldsymbol{Q}$ remain relevant during training? We leverage results from Fort et al. (2020), stating that the NTK (the kernel associated with the Jacobian feature map, see Section 3.2) changes significantly at the beginning of training and that its evolution slows down as

training goes on. This suggests that as a heuristic, we can compute the FIM-based directions after partial training, as they are unlikely to deviate much after the initial training. For this reason, UNLIMITD-F (Algorithm 2) first calls UNLIMITD-I (Algorithm 1) before computing the FIM-based $Q$ that yields intermediate parameters $\theta_0$ and $\mu$. Then the usual training takes place with the learning of $s$ in addition to $\theta_0$ and $\mu$.

## 5 RELATED WORK

**Bayesian inference with linearized DNNs.** Bayesian inference with neural networks is often intractable because the posterior predictive has rarely a closed-form expression. Whereas UNLIMITD linearizes the network to allow for practical Bayesian inference, existing work has used other approximations to tractably express the posterior. For example, it has been shown that in the infinite-width approximation, the posterior predictive of a Bayesian neural network behaves like a GP (Neal, 1996; Lee et al., 2017). This analysis can in some cases yield a good approximation to the Bayesian posterior of a DNN (Garriga-Alonso et al., 2018). It is also common to use Laplace's method to approximate the posterior predictive by a Gaussian distribution and allow practical use of the Bayesian framework for neural networks. This approximation relies in particular on the computation of the Hessian of the network: this is in general intractable, and most approaches use the so-called Gauss-Newton approximation of the Hessian instead (Ritter et al., 2018). Recently, it has been shown that the Laplace method using the Gauss-Newton approximation is equivalent to working with a certain linearized version of the network and its resulting posterior GP (Immer et al., 2021).

Bayesian inference is applied in a wide range of subjects. For example, recent advances in transfer learning have been possible thanks to Bayesian inference with linearized neural networks. Maddox et al. (2021) have linearized pre-trained networks and performed domain adaptation by conditioning the prior predictive with data from the new task: the posterior predictive is then used to make predictions. Our approach leverages a similar adaption method and demonstrates how the prior distribution can be learned in a meta-learning setup.

**Meta-learning.** MAML is a meta-learning algorithm that uses as adaptation a few steps of gradient descent (Finn et al., 2017). It has the benefit of being model-agnostic (it can be used on any model for which we can compute gradients w.r.t. the weights), whereas UNLIMITD requires the model to be a differentiable regressor. MAML has been further generalized to probabilistic meta-learning models such as PLATIPUS or BaMAML (Yoon et al., 2018; Finn et al., 2018), where the simple gradient descent step is augmented to perform approximate Bayesian inference. These approaches, like ours, learn (during meta-training) and make use of (at test-time) a prior distribution on the weights. In contrast, however, UNLIMITD uses exact Bayesian inference at test-time. MAML has also been improved for multimodal meta-learning via MMAML (Vuorio et al., 2019; Abdollahzadeh et al., 2021). Similarly to our method, they add a step to identify the cluster from which the task comes from (Vuorio et al., 2019). OoD detection in meta-learning has been studied by Jeong & Kim (2020), who build upon MAML to perform OoD detection in the classification setting, to identify unseen classes during training. Iwata & Kumagai (2022) also implemented OoD detection for classification, by learning a Gaussian mixture model on a latent space. UNLIMITD extends these ideas to the regression task, aiming to identify when test data is drawn from an unfamiliar function.

ALPaCA is a Bayesian meta-learning algorithm for neural networks, where only the last layer is Bayesian (Harrison et al., 2018). Such framework yields an exact linear regression that uses as feature map the activations right before the last layer. Our work is a generalization of ALPaCA, in the sense that UNLIMITD restricted to the last layer matches ALPaCA's approach. More on this link between the methods is discussed in Appendix A.2.

## 6 RESULTS AND DISCUSSION

We wish to evaluate four key aspects of UNLIMITD. (1) At test time, how do the probabilistic predictions compare to baselines? (2) How well does the detection of context data from OoD tasks perform? (3) How do these results hold in the multimodal setting? (4) Which approach performs better between (a) the identity covariance (UNLIMITD-I), (b) the low-dimensional covariance with random directions (UNLIMITD-R) and the compromise (c) using FIM-based directions (UNLIMITD-F) (see trade-off in Section 4.1.1)? That is, what is best between learning a rich prior distribution over the weights, keeping a full feature map, and a compromise between the two?

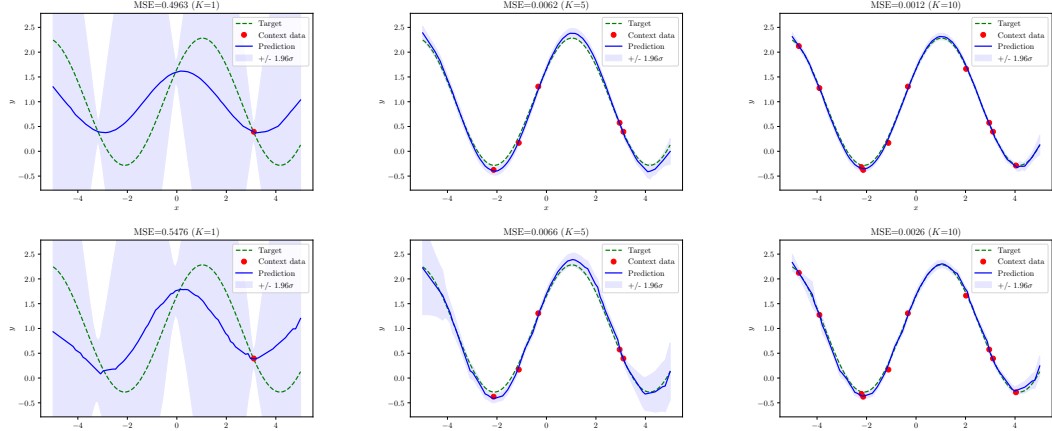

Figure 2: Example of predictions for a varying number of context inputs $K$, after meta-training with UNLIMITD-F. Top: UNLIMITD-F, infinite task dataset. Bottom: UNLIMITD-F, finite task dataset. The standard deviation is from the posterior predictive distribution. Note how the uncertainty levels are coherent with the actual prediction error. Also, note how uncertainty decreases when there is more context data. Notice how UNLIMITD-F recovers the shape of the sine even with a low number of context inputs. Finally, note how UNLIMITD-F is able to reconstruct the sine even when trained on fewer tasks (bottom). More comprehensive plots available in Figure 6.

We consider a cluster of sine tasks, one of linear tasks and one of quadratic tasks, regression problems inspired from Vuorio et al. (2019). Details on the problems can be found in Appendix A.4.

**Unimodal meta-learning: The meta-learned prior accurately fits the tasks.** First, we investigate the performance of UNLIMITD on a unimodal task distribution consisting of sinusoids of varying amplitude and phase, using the single GP structure for $\tilde{p}_\xi(f)$. We compare the performance between UNLIMITD-I, UNLIMITD-R and UNLIMITD-F. We also compare the results between training with an infinite amount of available sine tasks (infinite task dataset), and with a finite amount of available tasks (finite task dataset). More training details can be found in Appendix A.5. Examples of predictions at the test time are available in Figure 2, along with confidence levels.

In both OoD detection and quality of predictions, UNLIMITD-R and UNLIMITD-F perform better than UNLIMITD-I (Figure 3), and this is reflected in the quality of the learned prior $\tilde{p}_\xi(f)$ in each case (see Appendix A.3.1). With respect to the trade-off mentioned in Section 4.1.1, we find that for small networks, a rich prior over the weights matters more than the full expressiveness of the feature map, making both UNLIMITD-R and UNLIMITD-F appealing. However, after running further experiments on a deep-learning image-domain problem, this conclusion does not hold for deep networks (see Appendix A.3.3), where keeping an expressive feature map is important (UNLIMITD-I and UNLIMITD-F are appealing in that case). Thus, UNLIMITD-F is the variant that we retain, for it allows similar or better performances than the other variants in all situations.

Note how UNLIMITD-F outperforms MAML: it achieves much better generalization when decreasing the number of context samples $K$ (Figure 3). Indeed, UNLIMITD-F trained with a finite task dataset performs better than MAML with an infinite task dataset: it is able to capture better the common features of the tasks with a smaller task dataset.

**Multimodal meta-learning: Comparing the mixture model against a single GP.** Next, we consider a multimodal task distribution with training data consisting of sinusoids as well as lines with varying slopes. Here, we compare the performance between choosing the mixture structure or the single GP structure (see discussion in Section 4.1.2): in both cases, we use UNLIMITD-F. More training details can be found in Appendix A.6.

Both the OoD detection and the prediction performances are better with the mixture structure than with the single GP structure (Figure 4), indicating that the mixture model is a useful structure for $\tilde{p}_\xi(f)$. This is reflected in the quality of the learned priors (see Appendix A.3.2 for qualitative results including samples from the learned priors). Note how the single GP structure still performs better than both MAML and MMAML for prediction, especially in the low-data regime. This demonstrates

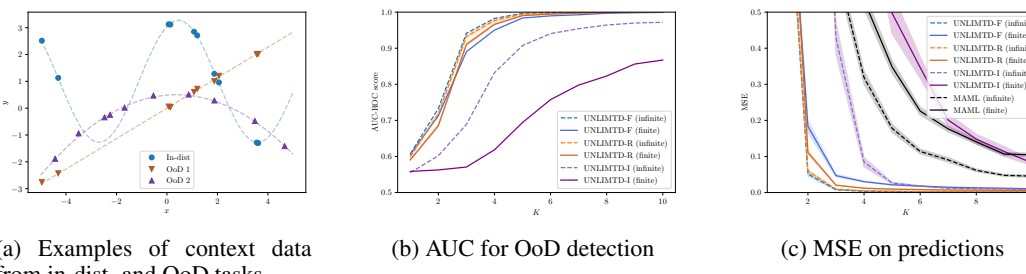

(a) Examples of context data from in-dist. and OoD tasks

(b) AUC for OoD detection

(c) MSE on predictions

Figure 3: Unimodal case: Performance of UNLIMITD for OoD detection and inference, as a function of the number of context datapoints $K$. The training dataset consists of sinusoids, while OoD tasks are lines and quadratic tasks. We compare different variants (UNLIMITD-I, UNLIMITD-R and UNLIMITD-F), and against MAML for predictions. We also compare training with a finite and infinite task dataset. Note how UNLIMITD-R and UNLIMITD-F have efficient OoD detection and outperform MAML in predictions. Also, note how MAML trained with an infinite task dataset performs worse than UNLIMITD-R and UNLIMITD-F trained on a finite task dataset.

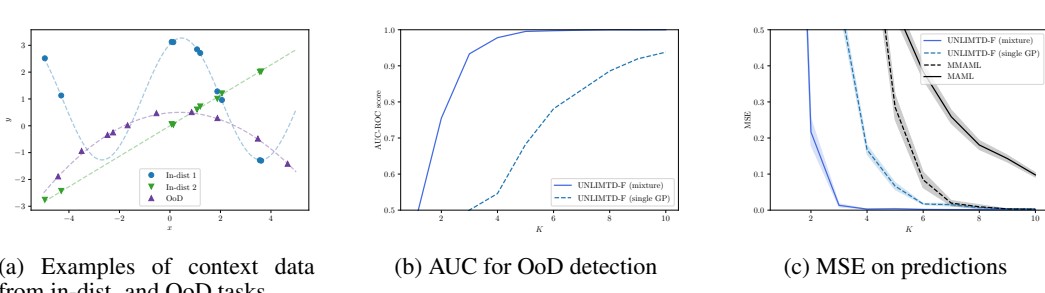

(a) Examples of context data from in-dist. and OoD tasks

(b) AUC for OoD detection

(c) MSE on predictions

Figure 4: Multimodal case: Performance of UNLIMITD for OoD detection and inference, as a function of the number of context datapoints $K$. The training dataset includes both sines and lines, while OoD tasks are quadratic functions. We compare the different variants (UNLIMITD-F with a single GP or a mixture model), and against MAML/MMAML for predictions. Note how both versions of UNLIMITD-F yield better predictions than the baselines. In particular, even with a single GP, UNLIMITD-F outperforms the baselines.

the strength of our probabilistic approach for multimodal meta-learning: even if the probabilistic assumptions are not optimal, the predictions are still accurate and can beat baselines.

## 7 CONCLUSION

We propose UNLIMITD, a novel meta-learning algorithm that models the underlying task distribution using a parametric and tuneable distribution, leveraging Bayesian inference with linearized neural networks. We compare three variants, and show that among these, the Fisher based parameterization, UNLIMITD-F, effectively balances scalability and expressivity, even for deep learning applications. We have demonstrated that (1) our approach makes efficient probabilistic predictions on in-distribution tasks, that compare favorably to, and often outperform, baselines, (2) it allows for effective detection of context data from OoD tasks and (3) that both these findings continue to hold in the multimodal task-distribution setting.

There are several avenues for future work. One direction entails understanding how the performance of UNLIMITD-F is impacted if the FIM-based directions are computed too early in training and the NTK changes significantly afterwards. One could also generalize our approach to non-Gaussian likelihoods, making UNLIMITD effective on classification tasks. Finally, further research can push the limits of multimodal meta-learning, e.g. by implementing non-parametric Bayesian methods to automatically infer an optimal number of clusters, thereby eliminating a hyperparameter of the current approach.

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

# A APPENDIX

## A.1 A TRACTABLE WAY OF FINDING THE PERTURBATION DIRECTIONS IN WEIGHT SPACE THAT IMPACT THE MOST THE PREDICTIONS OF AN ENTIRE DATASET

Deep neural networks have a large number of parameters, making the feature map $\phi_{\boldsymbol{\theta}_0}$ high-dimensional. However, recent work has shown that only a small subspace of the weight space is impactful. For example, to perform continual learning, Farajtabar et al. (2020) leverage the fact that it is sufficient to update the parameters orthogonally to a few directions only to avoid catastrophic forgetting. Sagun et al. (2017) have shown that the Hessian of a deep neural network can be summarized in a few number of directions, due to rapid spectral decay. This encourages finding a method to extract these meaningful directions of the weight space.

### A.1.1 LINK WITH THE FISHER INFORMATION MATRIX

We define these main directions as the ones that have the most impact on the predictions of a whole dataset. To find them, we first find a way to quantify the influence of an infinitesimal weight perturbation. Using the second-order approximation of that quantity, we then describe in the deep-learning context a tractable way to find the directions.

**Setting.** We take the same setting as Section 3.2, and we describe a method to quantify the influence of a parameter perturbation $\tilde{\boldsymbol{\theta}}$ on the predictions of a dataset of tasks $\mathcal{D}$. To do so, we leverage a probabilistic interpretation of the model: we assume a Gaussian pdf over the observations for given inputs and parameters $p_{\boldsymbol{\theta}}(\boldsymbol{Y}|\boldsymbol{X}) \sim \mathcal{N}(g(\boldsymbol{\theta}, \boldsymbol{X}), \boldsymbol{\Sigma}_\varepsilon)$, where the covariance of the noise $\boldsymbol{\Sigma}_\varepsilon$ is diagonal $\boldsymbol{\Sigma}_\varepsilon = \sigma_\varepsilon^2 \boldsymbol{I}$ (just as in Section 3.1).

**Perturbation of the prediction of a batch of inputs.** Before quantifying the influence of the perturbation on the predictions of the whole task dataset, we do it for the prediction of a batch of inputs $g(\boldsymbol{\theta}_0, \boldsymbol{X})$.

We borrow the method from Sharma et al. (2021): we quantify the influence of a parameter perturbation by computing the Kullback-Leibler divergence between $p_{\boldsymbol{\theta}_0}(\boldsymbol{Y}|\boldsymbol{X})$ and $p_{\boldsymbol{\theta}_0+\tilde{\boldsymbol{\theta}}}(\boldsymbol{Y}|\boldsymbol{X})$. The expansion is:

$$\delta(\boldsymbol{\theta}_0, \boldsymbol{X})(\tilde{\boldsymbol{\theta}}) := D_{\mathrm{KL}}(p_{\boldsymbol{\theta}_0}(\boldsymbol{Y}|\boldsymbol{X}) \| p_{\boldsymbol{\theta}_0+\tilde{\boldsymbol{\theta}}}(\boldsymbol{Y}|\boldsymbol{X})) \approx \tilde{\boldsymbol{\theta}}^\top \boldsymbol{F}(\boldsymbol{\theta}_0, \boldsymbol{X})\tilde{\boldsymbol{\theta}} + o(\|\tilde{\boldsymbol{\theta}}\|^2) \tag{7}$$

where $\boldsymbol{F}(\boldsymbol{\theta}_0, \boldsymbol{X}) := \boldsymbol{J}(\boldsymbol{\theta}_0, \boldsymbol{X})^\top \boldsymbol{\Sigma}_\varepsilon^{-1} \boldsymbol{J}(\boldsymbol{\theta}_0, \boldsymbol{X}) = \sigma_\varepsilon^{-2} \boldsymbol{J}(\boldsymbol{\theta}_0, \boldsymbol{X})^\top \boldsymbol{J}(\boldsymbol{\theta}_0, \boldsymbol{X}) \in \mathbb{R}^{P \times P}$ is the empirical Fisher Information Matrix (FIM) of the batch of inputs $\boldsymbol{X}$, computed on the parameters $\boldsymbol{\theta}_0$.

**Generalization: perturbation of the prediction of a dataset.** Now we can define the influence of a parameter perturbation on the whole training dataset $\mathcal{D}$, by generalizing the previous definition:

$$\delta(\boldsymbol{\theta}_0, \mathcal{D})(\tilde{\boldsymbol{\theta}}) := \frac{1}{N} \sum_{i=1}^{N} \delta(\boldsymbol{\theta}_0, \widetilde{\boldsymbol{X}^i})(\tilde{\boldsymbol{\theta}})$$

Using equation 7, this quantity verifies:

$$\delta(\boldsymbol{\theta}_0, \mathcal{D})(\tilde{\boldsymbol{\theta}}) \approx \tilde{\boldsymbol{\theta}}^\top \left( \frac{1}{N} \sum_{i=1}^N \boldsymbol{F}(\boldsymbol{\theta}_0, \widetilde{\boldsymbol{X}^i}) \right) \tilde{\boldsymbol{\theta}} + o(\|\tilde{\boldsymbol{\theta}}\|^2) \tag{8}$$

which gives a natural definition for the FIM of the whole dataset by analogy with equation 7:

$$\boldsymbol{F}(\boldsymbol{\theta}_0, \mathcal{D}) := \frac{1}{N} \sum_{i=1}^N \boldsymbol{F}(\boldsymbol{\theta}_0, \widetilde{\boldsymbol{X}^i}) = \frac{1}{N\sigma_\varepsilon^2} \sum_{i=1}^N \boldsymbol{J}(\boldsymbol{\theta}_0, \widetilde{\boldsymbol{X}^i})^\top \boldsymbol{J}(\boldsymbol{\theta}_0, \widetilde{\boldsymbol{X}^i}) \in \mathbb{R}^{P \times P}$$

The expansion in equation 8 shows that the FIM of the dataset is a second-order approximation describing the influence of a parameter perturbation over the entire dataset. In particular, the eigenvectors of $\boldsymbol{F}(\boldsymbol{\theta}_0, \mathcal{D})$ with the highest eigenvalues are the directions that impact the most the predictions.

### A.1.2 COMPUTING THE TOP EIGENVECTORS OF THE FISHER INFORMATION MATRIX OF THE DATASET IN A DEEP LEARNING CONTEXT

Naively computing the top eigenspace of $\boldsymbol{F}(\boldsymbol{\theta}_0, \mathcal{D})$ requires processing a $P \times P$ matrix, which is intractable in the deep-learning context (where $P$ can surpass $10^6$). Instead, we decide to use the method used by Sharma et al. (2021), which leverages a low-rank-approximation-based technique (namely matrix sketching) developed by Tropp et al. (2017).

**Sketching the FIM of the dataset.** The key idea behind this technique is to build two small random sketches of the FIM, $(\boldsymbol{Y}, \boldsymbol{W}) := \mathcal{S}(\boldsymbol{F}(\boldsymbol{\theta}_0, \mathcal{D}))$, that with high probability contain enough information to reconstruct the top $s$ eigenvectors of $\boldsymbol{F}(\boldsymbol{\theta}_0, \mathcal{D})$. The linearity of $\mathcal{S}$ simplifies the sketching process by breaking down the computation into individual sketches:

$$(\boldsymbol{Y}, \boldsymbol{W}) := \mathcal{S}(\boldsymbol{F}(\boldsymbol{\theta}_0, \mathcal{D})) = \frac{1}{N\sigma_\varepsilon^2} \sum_{i=1}^N \mathcal{S}\left( \boldsymbol{J}(\boldsymbol{\theta}_0, \widetilde{\boldsymbol{X}^i})^\top \boldsymbol{J}(\boldsymbol{\theta}_0, \widetilde{\boldsymbol{X}^i}) \right) =: \frac{1}{N\sigma_\varepsilon^2} \sum_{i=1}^N (\boldsymbol{Y}^i, \boldsymbol{W}^i)$$

In particular, the sketch $(\boldsymbol{Y}, \boldsymbol{W})$ can be updated in-place and does not require to store all the individual sketches. Given a sketch budget $k + l$ (Sharma et al. (2021) recommends choosing $k := 2s + 1$ and $l := 4s + 3$) and two random normal matrices $\boldsymbol{\Omega} \in \mathbb{R}^{k \times P}$ and $\boldsymbol{\Psi} \in \mathbb{R}^{l \times P}$, the random individual sketches $(\boldsymbol{W}^i, \boldsymbol{Y}^i)$ are defined as:

$$\begin{cases} \boldsymbol{Y}^i & := & ((\boldsymbol{\Omega}\boldsymbol{J}_i^\top)\boldsymbol{J}_i)^\top & \in \mathbb{R}^{P \times k} \\ \boldsymbol{W}^i & := & (\boldsymbol{\Psi}\boldsymbol{J}_i^\top)\boldsymbol{J}_i & \in \mathbb{R}^{l \times P} \end{cases}$$

where $\boldsymbol{J}_i := \boldsymbol{J}(\boldsymbol{\theta}_0, \widetilde{\boldsymbol{X}^i})$.

**Sketch-based computation of the top eigenspace.** Once the sketches are computed, the function FIXEDRANKSYMAPPROX by Tropp et al. (2017) computes the first eigenvectors and eigenvalues of the FIM of the dataset. Overall, the memory footprint to find the sketches and the top eigenspace is $O(P(s + N_y M))$, where $s$ is the number of queried eigenvectors and $M$ is the size of $\widetilde{\boldsymbol{X}^i}$. As long as $s \ll P$, this computation is tractable, given that $N_y M \ll P$ is usual deep learning contexts. Algorithm 3 summarizes the process that yields the FIM-based projections via sketching. We drop the scaling coefficient $\sigma_\varepsilon^{-2}$ as it doesn't affect the computation, given that we only want orthogonal eigenvectors and eigenvalues.

### A.2 UNLIMITD AS A GENERALIZATION OF ALPACA

**Restraining the linearization to the last layer.** Remember the linear regression of equation 3, that we obtained by linearizing the network with all its layers. Let's separate the parameters of the network $\boldsymbol{\theta}$ into two: the parameters of all the layers but the last one $\boldsymbol{\lambda}$, and the parameters of the last layer $\boldsymbol{\rho}$: $\boldsymbol{\theta} = \boldsymbol{\lambda} \cup \boldsymbol{\rho}$.

We assume that the last layer is dense with biases: we will note $\boldsymbol{\rho}^w$ the weight matrix and $\boldsymbol{\rho}^b$ the biases of this last layer. $N_\psi$ will stand as the dimension of the activations right before last

---

**Algorithm 3** Computing the FIM-based projections

---

**Require:** $s$ (desired size of the subspace)
1: $k \leftarrow 2s + 1$
2: $l \leftarrow 4s + 3$
3: Draw $\boldsymbol{\Omega} \in \mathbb{R}^{k \times P}, \boldsymbol{\Psi} \in \mathbb{R}^{l \times P}$, two random normal matrices
4: Initialize $\boldsymbol{Y} = 0 \in \mathbb{R}^{P \times k}, \boldsymbol{W} = 0 \in \mathbb{R}^{l \times P}$
5: **for all** training task $\mathcal{T}^i$ **do**
6: $\quad \boldsymbol{J}_i \leftarrow \boldsymbol{J}(\boldsymbol{\theta}_0, \widetilde{\boldsymbol{X}^i})$
7: $\quad \boldsymbol{Y} \leftarrow \boldsymbol{Y} + 1/N((\boldsymbol{\Omega} \boldsymbol{J}_i^\top) \boldsymbol{J}_i)^\top$
8: $\quad \boldsymbol{W} \leftarrow \boldsymbol{W} + 1/N(\boldsymbol{\Psi} \boldsymbol{J}_i^\top) \boldsymbol{J}_i$
9: **end for**

---

layer: in particular, $\boldsymbol{\rho}^w \in \mathbb{R}^{N_y \times N_\psi}$ and $\boldsymbol{\rho}^b \in \mathbb{R}^{N_y}$. $P'$ will stand as the size of $\boldsymbol{\rho}$: in our case, $P' = N_\psi \times N_y + N_y$. We implicitly vectorize $\boldsymbol{\rho}$, such that:

$$\boldsymbol{\rho} = \begin{pmatrix} \text{vec } \boldsymbol{\rho}^w \\ \boldsymbol{\rho}^b \end{pmatrix} \in \mathbb{R}^{N_\psi N_y + N_y} = \mathbb{R}^{P'}$$

We now restrain the linear regression to the last layer as follows:

$$\boldsymbol{y} = \boldsymbol{J}'(\boldsymbol{\rho}_0, \psi_{\boldsymbol{\lambda}}(\boldsymbol{X}))(\boldsymbol{\rho} - \boldsymbol{\rho}_0) + \varepsilon \tag{9}$$

where:

- $\psi_{\boldsymbol{\lambda}}(\cdot) : \mathbb{R}^{N_x \times K} \to \mathbb{R}^{N_\psi \times K}$ stands for the function that maps the inputs and the activations right before the last layer;

- $\boldsymbol{J}'(\cdot, \cdot) : \mathbb{R}^{P'} \times \mathbb{R}^{N_\psi \times K} \to \mathbb{R}^{N_y K \times P'}$ stands for the jacobian of the last layer *with respect to the parameters $\boldsymbol{\rho}$*. We can write the jacobian in closed-form due to the linearity of the last layer:

$$\boldsymbol{J}'(\boldsymbol{\rho}_0, \psi_{\boldsymbol{\lambda}}(\boldsymbol{X})) = \begin{pmatrix} \psi_{\boldsymbol{\lambda}}(\boldsymbol{X})^\top \otimes \boldsymbol{I}_{N_y} & \boldsymbol{I}_{N_y} \end{pmatrix} \in \mathbb{R}^{N_y K \times P'}$$

  Note that $\boldsymbol{J}'(\boldsymbol{\rho}_0, \psi_{\boldsymbol{\lambda}}(x))$ does not depend on the linearization point $\boldsymbol{\rho}_0$ (could be expected, given the linearity of the last layer).

- $\boldsymbol{\rho} - \boldsymbol{\rho}_0 \in \mathbb{R}^{P'}$ is the *correction* to the last parameters. We will note the correction to the weights as $\hat{\boldsymbol{\rho}}^w := \boldsymbol{\rho}^w - \boldsymbol{\rho}_0^w$ and the correction to the bias as $\hat{\boldsymbol{\rho}}^b := \boldsymbol{\rho}^b - \boldsymbol{\rho}_0^b$.

Using Kronecker's product identities, the linear regression in equation 9 can be rewritten:

$$\boldsymbol{y} = \hat{\boldsymbol{\rho}}^w \psi_{\boldsymbol{\lambda}}(\boldsymbol{X}) + \hat{\boldsymbol{\rho}}^b + \varepsilon \tag{10}$$

Also, as a side note, another way of getting this linear regression (equation 10) is to rewrite the initial linearization of Section 3.2, but with respect to $\boldsymbol{\rho}$ only:

$$
\begin{aligned}
f(\boldsymbol{\lambda} \cup \boldsymbol{\rho}, \boldsymbol{x}_t) - f(\boldsymbol{\lambda} \cup \boldsymbol{\rho}_0, \boldsymbol{x}_t) &= \boldsymbol{\rho}^w \psi_{\boldsymbol{\lambda}}(x) + \boldsymbol{\rho}^b - (\boldsymbol{\rho}_0^w \psi_{\boldsymbol{\lambda}}(x) + \boldsymbol{\rho}_0^b) \\
&= (\boldsymbol{\rho}^w - \boldsymbol{\rho}_0^w) \psi_{\boldsymbol{\lambda}}(x) + (\boldsymbol{\rho}^b - \boldsymbol{\rho}_0^b) \\
&= \hat{\boldsymbol{\rho}}^w \psi_{\boldsymbol{\lambda}}(x) + \hat{\boldsymbol{\rho}}^b
\end{aligned}
$$

Doing it this way has the benefit to show that the linearization is exact when restricted to the last layer. For non-linear neural networks, the linearization is always an approximation.

**Adapting UNLIMITD to the last layer.** Just like what we did in the general case, we use the GP theory to make Bayesian inference on this linear regression. The new parameters of the GP $\tilde{p}_\xi(f)$ are now $\xi = (\boldsymbol{\lambda}, \boldsymbol{\mu}, \boldsymbol{\Sigma})$, where $\boldsymbol{\mu}$ and $\boldsymbol{\Sigma}$ are the parameters of the Gaussian prior over $\hat{\boldsymbol{\rho}}$. Note how $\boldsymbol{\rho}_0$ have disappeared from $\xi$: contrary to the general case, the linearization point is not optimized, as it does not impact the computation. Also note that $\boldsymbol{\lambda}$ has replaced $\boldsymbol{\rho}_0$, as it parameterizes the feature map.

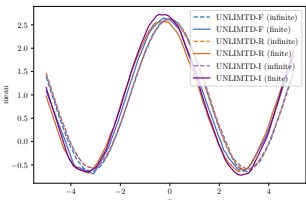 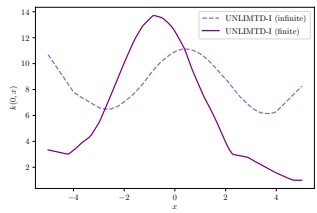 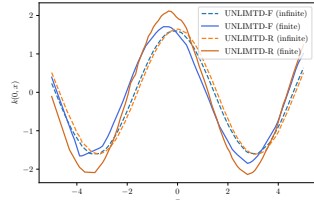

(a) Mean $\mathbb{E}(f)$ (UNLIMITD-I, UNLIMITD-R, UNLIMITD-F)

(b) Covariance $\mathrm{cov}(f(0), f(x))$ (UNLIMITD-I)

(c) Covariance $\mathrm{cov}(f(0), f(x))$ (UNLIMITD-R, UNLIMITD-F)

Figure 5: Mean (5a) and covariance (5b, 5c) functions of $\tilde{p}_\xi(f)$ after different meta-trainings on the sine cluster (UNLIMITD-I, UNLIMITD-R and UNLIMITD-F, with a finite or infinite dataset). Note the scale, and how UNLIMITD-I has a less accurate covariance function that UNLIMITD-R and UNLIMITD-F.

**Comparison with ALPaCA.** In the ALPaCA setting, the linear regression is:

$$\boldsymbol{y}_a = \hat{\boldsymbol{\rho}}^w \psi_{\boldsymbol{\lambda}}(\boldsymbol{X}) + \varepsilon \tag{11}$$

Note that $\hat{\boldsymbol{\rho}}^w$ still plays the same role in the linear regression, but is not any *correction* anymore. Substracting 11 - 10 yields:

$$\boldsymbol{y} - \boldsymbol{y}_a = \hat{\boldsymbol{\rho}}^b$$

UNLIMITD restricted to the last layer is closely related to ALPaCA, in that they both perform a linear regression with the same kernel: the only difference lies in the additional bias term that is not considered in ALPaCA. Thus, we can think of UNLIMITD as a generalization of ALPaCA to all the layers of the network.

## A.3 ADDITIONAL RESULTS

### A.3.1 ADDITIONAL RESULTS (SINGLE-CLUSTER CASE)

**Quality of the priors.** To qualitatively analyze the prior after meta-training on the sines, we plot the mean functions and the covariance functions of the resulting GP, that is $\tilde{p}_\xi(f)$ (Figure 5). All the trainings yield a similar mean for $\tilde{p}_\xi(f)$ (that is a cosine with amplitude 1.5 and offset 1) (Figure 5a), which is close to the theoretical mean of $p(f)$ (a cosine with amplitude 2.5 and offset 1). The covariance function of UNLIMITD-R and UNLIMITD-F (Figure 5c) resembles what we would expect for $p(f)$ (e.g. periodicity, negative correlation between 0 and $\pi$, etc.), but it is not the case for UNLIMITD-I (Figure 5b). This empirical analysis confirms the quantitative comparison in terms of OoD detection and prediction performance carried in Section 6.

**Examples of predictions.** Figure 6 summarizes the predictions of the model meta-trained on sines, for a varying number of context inputs $K$, breaking down all the different cases of training.

### A.3.2 ADDITIONAL RESULTS (MULTI-CLUSTER CASE)

**Quality of the priors.** Figure 7 and Figure 8 show the mean and covariance functions of the GP (when training with a single GP), and the mean and covariance functions of the GPs composing the mixture(when training with a mixture of GPs). We note that in the case of the mixture, both of the Gaussians composing the mixture have correctly captured the common features shared by each of the clusters (respectively the linear and the sine cluster): e.g. the mean of the line cluster is the zero-function, which matches Figure 7b, the correlation between $x = 1$ and the other inputs is correctly rendered for linear tasks (Figure 8b), etc.

When learning with a single GP, the learnt mean and covariance do not match any of the two clusters. For example, the mean has an intermediate offset between the offset of the sine cluster and the line cluster. This empirical analysis comforts the conclusions of Section 6: the mixture model yields better results than the single GP case.

### A.3.3 UNLIMITD YIELDS EFFECTIVE PREDICTIONS ON LARGE-SCALE VISION PROBLEMS

We consider a regression-vision meta-learning problem from Gao et al. (2022), Shapenet1D, aiming to predict object orientations in space. In this problem, each task consists of a different object of which we want to predict the orientation. For each task, the context data consists of some images of the same object, but with different orientations; the query inputs are other images of the same object, with unknown orientations. Details on the problems and the datasets can be found in appendix A.4.

We train a deep learning model on Shapenet1D, and we compare the performances on the test set between UNLIMITD-I, UNLIMITD-R and UNLIMITD-F (Figure 9). More training and test details can be found in the Appendix A.7.

Both UNLIMITD-I and UNLIMITD-F yield better performances than MAML, achieving low angle errors. UNLIMITD-R however gives poor results, worse than that of MAML.

In terms of the trade-off of Section 4.1.1, our conclusion from small networks does not scale up to deep models. Here, the loss of valuable features is perceptible (that is what happens with UNLIMITD-R, when the randomness of the directions may drop such features), and not learning a rich prior over the weights is not burdensome (UNLIMITD-I). However, UNLIMITD-F plays a role of compromise, by learning the prior covariance while keeping the few important features of the jacobian, as it gives comparable results to UNLIMITD-I.

## A.4 DETAILS ON THE REGRESSION PROBLEMS

### A.4.1 SIMPLE REGRESSION PROBLEMS

Our simple regression problems are inspired by Vuorio et al. (2019)'s work. They consists of three clusters of different types of tasks, and varying offset. The first cluster consists of sines with constant frequency and offset, but with a varying amplitude and phase:

$$\{x \mapsto A\sin(x + \varphi) + 1 | A \in [0.1, 5], \varphi \in [0, \pi]\}$$

The second cluster consists of lines with a varying slope and no offset:

$$\{x \mapsto ax | a \in [-1, 1]\}$$

The last cluster consists of quadratic functions, with a varying quadratic coefficient and phase, and with a constant offset:

$$\{x \mapsto a(x - \varphi)^2 + 0.5 | a \in [-0.2, 0.2], \varphi \in [-2, 2]\}$$

In all these clusters, we add an artificial Gaussian noise on the context observations $\mathcal{N}(0, \ 0.05)$. The query datapoints remain noiseless, to remain coherent with the assumption from Section 3.1 borrowed from Rasmussen & Williams (2005).

### A.4.2 VISION PROBLEM

We consider the meta-learning vision, regression problem recently created by Gao et al. (2022) (namely Shapenet1D), that consists in estimating objects orientation from images. The objects have a vertical angular degree of freedom, and Figure 10 shows an example of such objects in different positions.

We use the same training and test datasets as Gao et al. (2022), with no kind of augmentation (in particular, no artificial noise on the ground-truth angles). In particular, we use the same intra-category (IC) evaluation dataset (that is, objects from the same categories as the objects used for training) and cross-category (CC) evaluation dataset (that is, objects from different categories as the ones used for training).

## A.5 TRAINING AND TEST DETAILS FOR THE SINGLE CLUSTER CASE

### A.5.1 TRAINING DETAILS OF UNLIMITD (SINGLE CLUSTER CASE)

The model is a neural network with 2 hidden layers, 40 neurons each, with a ReLU non-linearity. In our single-cluster experiment, the cluster is the sine cluster (see Appendix A.4).

In the case where there are an infinite number of available sine tasks during training (*ie* $N = +\infty$), the training is performed with $n = 24$ tasks per epoch, and at each epoch the context inputs are

randomly drawn from $[-5, 5]$ (which means that $M = +\infty$). In the case where we restrict the available tasks to a finite number, we randomly choose $N = 10$ tasks and $M = 50$ context datapoints per task before training (they are shared among all the "finite" trainings) and perform the trainings with $n = 6$ tasks per epoch.

For all the trainings, the number of context inputs seen during training is $K = 10$.

For all trainings, we train UNLIMITD on $60,000$ epochs. When dealing with the UNLIMITD-F case, we allow half of the epochs ($30,000$) to the training *before* finding the intermediate $\boldsymbol{\theta}_0$ and $\boldsymbol{\mu}$ (see Algorithm 2), and the other half *after*.

In the UNLIMITD-F case with infinite available tasks, a finite number of tasks is needed to compute the FIM: thus we build an artificial finite dataset of $N = 100$ and $M = P$ (arbitrary, but chosen so that $\boldsymbol{J}(\boldsymbol{\theta}_0, \widetilde{\boldsymbol{X}^i}) \boldsymbol{J}(\boldsymbol{\theta}_0, \widetilde{\boldsymbol{X}^i})^\top$ gets a chance to be full-rank *ie* contain as much information as possible), that is used only for that computation.

In the UNLIMITD-F and the UNLIMITD-R cases, the subspace size is $s = 10$.

For all trainings, the meta-optimizer is Adam with an initial learning rate of $0.001$. The noise $\sigma_\varepsilon = 0.05$, equal to the noise added to the context data. We compute the NLL using Rasmussen & Williams (2005)'s implementation.

### A.5.2 TRAINING DETAILS OF MAML (SINGLE CLUSTER CASE)

We train MAML baselines to compare our results with that of MAML. A large part of these hyper-parameters is directly inspired from Finn et al. (2017)'s work.

The model is a neural network with 2 hidden layers, 40 neurons each, with a ReLU non-linearity. In our single-cluster experiment, the cluster is the sine cluster (see Appendix A.4).

In the case where there are an infinite number of available sine tasks during training (*ie* $N = +\infty$), the training is performed with $n = 24$ tasks per epoch, and at each epoch the context and query inputs are randomly drawn from $[-5, 5]$. In the case where we restrict the available tasks to a finite number, we randomly choose $N = 10$ tasks and $M = 50$ datapoints per task (used for both context and query batches) before training (they are shared among all the "finite" trainings) and perform the trainings with $n = 6$ tasks per epoch.

For all trainings, the number of context datapoints is $K = 10$, and the number of query datapoints is $L = 10$. We meta-train for $70,000$ epochs.

For all trainings, the meta-optimizer is Adam with an initial learning rate of $0.001$. The inner learning-rate is kept constant, at $0.001$. The number of inner updates is 5 during training, and 10 at test time.

### A.5.3 TEST DETAILS (SINGLE-CLUSTER CASE)

For the OoD detection evaluation, we plot the AUC as a function of the number of the context inputs $K$. The AUC is computed using the NLL of the context inputs wrt to $\tilde{p}_\xi(f)$ (our uncertainty metric). The true-positives are the OoD tasks (lines and quadratic tasks) flagged as such; the false-positives are the in-distribution tasks (sines) flagged as OoD.

For the predictions, we plot the average and ci95 of the MSE on 1,000 tasks (100 queried inputs each). In the UNLIMITD-R case, we also compute the average and ci95 on 5 different random projections trainings.

### A.6 TRAINING AND TEST DETAILS FOR THE MULTI CLUSTER CASE

### A.6.1 TRAINING DETAILS OF UNLIMITD (MULTI-CLUSTER CASE)

The model is a neural network with 2 hidden layers, 40 neurons each, with a ReLU non-linearity. In our multi-cluster experiment, $\alpha = 2$: the clusters consist of the sine cluster and the linear cluster (see Appendix A.4).

For all the trainings, the training is performed with $n = 24$ tasks per epoch (with an infinite number of available sine tasks and linear tasks during training *ie* $N = +\infty$), and at each epoch the context inputs are randomly drawn from $[-5, 5]$ (which means that $M = +\infty$). In accordance with our

equal probability assumption from Section 4.1.2, at each epoch $n/2 = 12$ tasks come from the sine cluster and $n/2 = 12$ tasks come from the linear cluster.

For all trainings, the number of context inputs seen during training is $K = 10$.

For all trainings, we train UNLIMITD algorithm on $60,000$ epochs: because we deal with the UN-LIMITD-F case, we allow half of the epochs ($30,000$) to the training *before* finding the intermediate $\boldsymbol{\theta}_0$ and $\{\boldsymbol{\mu}_j\}_{j=1}^{j=\alpha}$ (see Algorithm 2), and the other half *after*.

In the UNLIMITD-F case, a finite number of tasks is needed to compute the FIM: thus we build an artificial finite dataset of $N = 100$ and $M = P$ (arbitrary, but chosen so that $\boldsymbol{J}(\boldsymbol{\theta}_0, \widetilde{\boldsymbol{X}}^i)\boldsymbol{J}(\boldsymbol{\theta}_0, \widetilde{\boldsymbol{X}}^i)^\top$ gets a chance to be full-rank *ie* contain as much information as possible), that is used only for that computation.

For all trainings, the subspace size is $s = 10$.

For all trainings, the meta-optimizer is Adam with an initial learning rate of $0.001$. The noise $\sigma_\varepsilon = 0.05$, equal to the noise added to the context data. We compute the NLL using Rasmussen & Williams (2005)'s implementation.

When training with MIXT, we make sure to initialize $\boldsymbol{s}_1$ and $\boldsymbol{s}_2$ randomly with $\mathcal{N}(\boldsymbol{0}, \ 0.5\boldsymbol{I})$, so that the meta-learning can effectively differentiate the two clusters. Also, in the MIXT case, the mean $\boldsymbol{\mu}$ is unique *before* computing the FIM. Once we have found the projection directions, we initialize $(\boldsymbol{\mu}_1, \boldsymbol{\mu}_2)$ with the intermediate $\boldsymbol{\mu}$, thus yielding two Gaussians.

### A.6.2 TRAINING DETAILS OF MAML (MULTI-CLUSTER CASE)

We reimplement and train a MAML baseline to compare our results with that of MAML. A large part of these hyperparameters is directly inspired from Finn et al. (2017)'s work.

The model is a neural network with 2 hidden layers, 40 neurons each, with a ReLU non-linearity. In our multi-cluster experiment, $\alpha = 2$: the clusters consist of the sine cluster and the linear cluster (see Appendix A.4).

The training is performed with $n = 24$ tasks per epoch (with an infinite number of available sine and linear tasks during training *ie* $N = +\infty$) and at each epoch the context and query inputs are randomly drawn from $[-5, 5]$. In accordance with our equal probability assumption from Section 4.1.2, at each epoch $n/2 = 12$ tasks come from the sine cluster and $n/2 = 12$ tasks come from the linear cluster.

For all trainings, the number of context datapoints is $K = 10$, and the number of query datapoints is $L = 10$. We meta-train for $70,000$ epochs.

For all trainings, the meta-optimizer is Adam with an initial learning rate of $0.001$. The inner learning-rate is kept constant, at $0.001$. The number of inner updates is 5 during training, and 10 at test time.

### A.6.3 TRAINING DETAILS OF MMAML (MULTI-CLUSTER CASE)

We train a MMAML by using Vuorio et al. (2019)'s code, using the best setting mentionned in their paper (FiLM). We adapt their code to train it on our sine and linear clusters: we add an offset to their sine cluster (via their parameter `bias`), change the phase from $\sin(x - \varphi)$ to $\sin(x + \varphi)$ and specify via the arguments the slope range ($[-1, 1]$) and the y-intercept range ($[0, 0]$, because it does not vary in our case) of the line tasks. We also change the number of context inputs that we set to $K = 10$, to remain coherent with the rest of the trainings. The rest of the hyperparameters are kept identical to the command specified in the repository of MMAML. Finally, at test time we make the query ground-truths noiseless, to remain coherent with the rest of the test conditions.

### A.6.4 TEST DETAILS (MULTI-CLUSTER CASE)

For the OoD detection evaluation, we plot the AUC as a function of the number of the context inputs $K$. The AUC is computed using the NLL of the context inputs wrt to $\tilde{p}_\xi(f)$ (our uncertainty metric). The true-positives are the OoD tasks (quadratic tasks) flagged as such; the false-positives are the in-distribution tasks (lines and sines) flagged as OoD.

For the predictions, we plot the average and ci95 of the MSE on 1,000 tasks (100 queried inputs each): half of them are sines and half of them are lines.

## A.7  Training and test details for the vision problem

The model is a deep neural network, identical to the one used by Gao et al. (2022) except for the last layer: instead of doing a one-dimensional regression (where the output stands for an angle prediction), we perform a two-dimensional regression (where the output stands for a cosine and sine prediction). This choice is motivated by the fact that the Gaussian noise assumed in Section 3.1 cannot capture the complexity of an angle error (e.g. predicting $361°$ should yield a low error when compared to the ground-truth angle $0°$), but better renders the MSE that can be applied on cosine and sine.

### A.7.1  Training details of UnLiMiTD (vision case)

For all the trainings, there are $n = 10$ tasks per epoch and $K = 15$ context inputs per task. When dealing with the UnLiMiTD-F case, we allow half of the epochs ($5,000$) to the training *before* finding the intermediate $\theta_0$ and $\mu$ (see Algorithm 2), and the other half after.

In the UnLiMiTD-F and UnLiMiTD-R cases, the subspace size is $s = 100$.

For all the trainings, the meta-optimizer is Adam with an initial learning rate of $0.001$. The noise is $\sigma_\varepsilon = 0.01$. We compute the NLL using Rasmussen & Williams (2005)'s implementation.

### A.7.2  Training details of MAML (vision case)

We train a MAML baseline to compare our results with that of MAML. A large part of these hyper-parameters is directly inspired from Gao et al. (2022).

The training is performed with $n = 10$ tasks per epoch. The number of context datapoints is $K = 15$, and the number of query datapoints is $L = 10$. We meta-train for $50,000$ epochs.

For all trainings, the meta-optimizer is Adam with an initial learning rate of $0.0005$. The inner learning-rate is kept constant, at $0.002$. The number of inner updates is 5 during training, and 20 a test time.

### A.7.3  Test details (vision problem)

At test time, we wrap our model with the $\mathrm{arctan}$ function to convert the predictions angle predictions. Then, we compare the angle predictions with the ground-truth angles. To do so, we use the following error from Gao et al. (2022) to compare two angles:

$$\mathcal{E}(\beta, \beta^*) = \min\{\mathcal{E}_{\beta^+,\beta^*}, \mathcal{E}_{\beta,\beta^*}, \mathcal{E}_{\beta^-,\beta^*}\}$$

where $\mathcal{E}_{\beta^\pm,\beta^*} = |y \pm 360 - y^*|$ and $\mathcal{E}_{\beta,\beta^*} = |y - y^*|$.

When plotting the performance (Figure 9), we plot the average and ci95 on 100 tasks (15 queried inputs each). For UnLiMiTD-R, the average and ci95 are computed on 5 different random projection trainings.

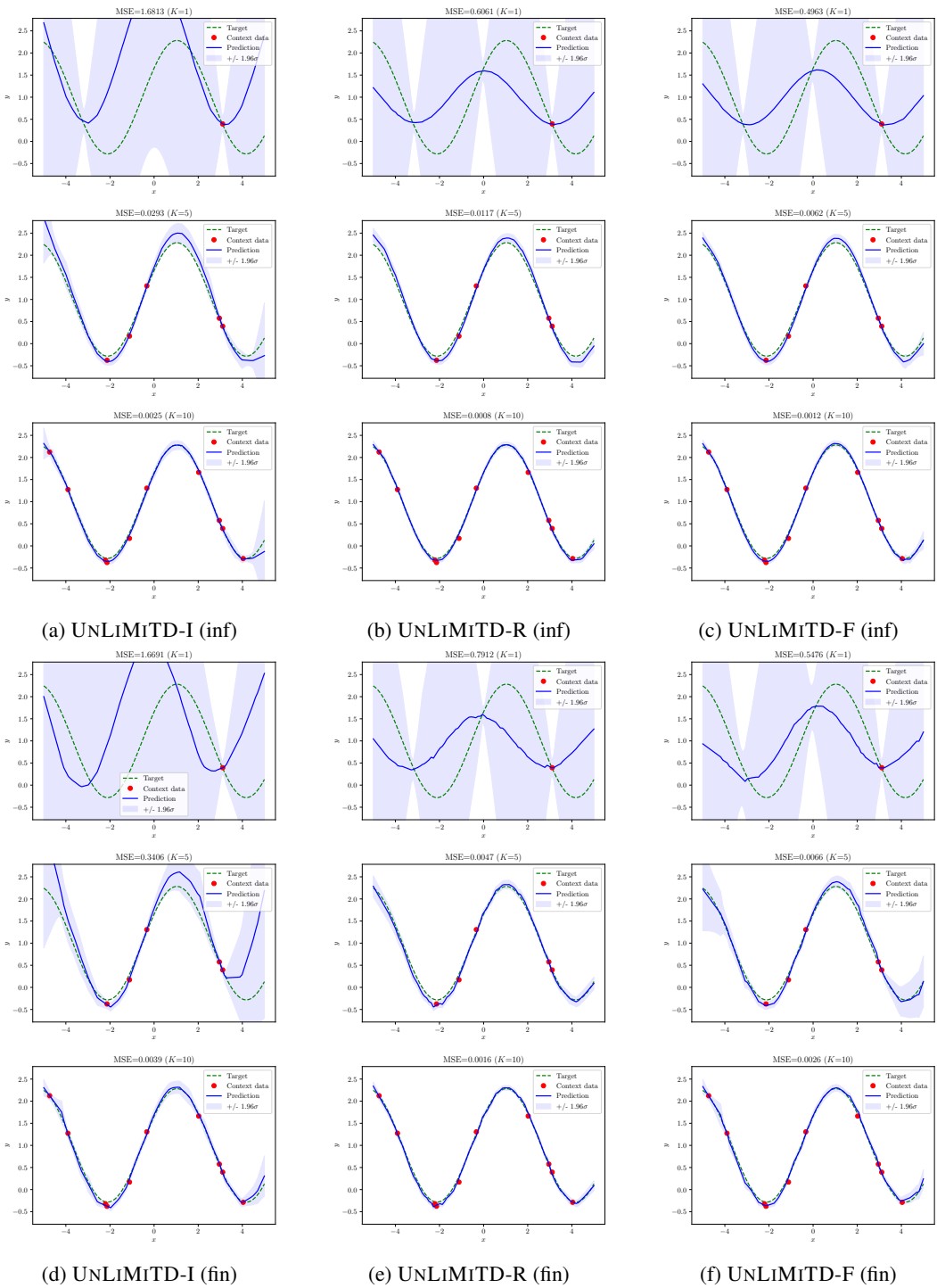

Figure 6: Example of predictions for a varying number of context inputs $K$, after different meta-trainings on the sine cluster (UNLIMITD-I, UNLIMITD-R and UNLIMITD-F, in the infinite and finite case). Standard deviation is from the posterior predictive distribution. Note how UNLIMITD-R and UNLIMITD-F perform better than UNLIMITD-I when it comes to reconstructing the sine with a smaller amount of context inputs.

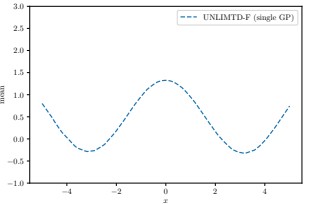
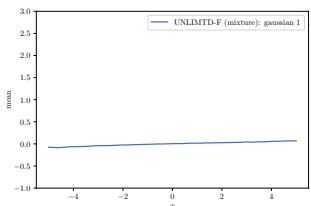
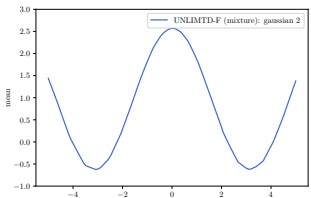

(a) Mean function $\mathbb{E}(f)$ of the GP (UNLIMITD-F with a single GP).

(b) Mean function $\mathbb{E}(f)$ of the first GP of the mixture (UNLIMITD-F with a mixture of GPs).

(c) Mean function $\mathbb{E}(f)$ of the second GP of the mixture (UNLIMITD-F with a mixture of GPs).

Figure 7: Mean functions of the GP (when learning with a GP) / the GPs composing the mixture (when learning a mixture of GPs), after training on both the sine and line cluster with UNLIMITD-F. Note how the mean of the single GP is intermediate between the ones of the mixture.

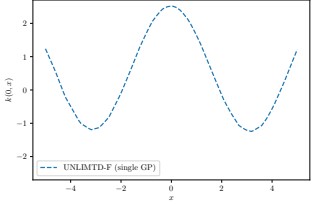
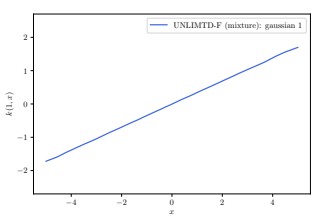
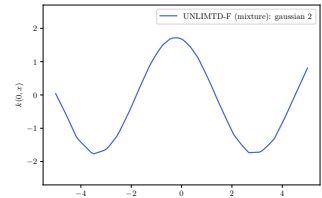

(a) Covariance function $\mathrm{cov}(f(0), f(x))$ of the GP (UNLIMITD-F with a single GP).

(b) Covariance function $\mathrm{cov}(f(1), f(x))$ of the first GP of the mixture (UNLIMITD-F with a mixture of GPs)

(c) Covariance function $\mathrm{cov}(f(0), f(x))$ of the second GP of the mixture (UNLIMITD-F with a mixture of GPs)

Figure 8: Covariance functions of the GP (when learning with a GP) / the GPs composing the mixture (when learning a mixture of GPs), after training on both the sine and line cluster with UNLIMITD-F. Note how the covariance of the single GP is not accurate for any of the two clusters.

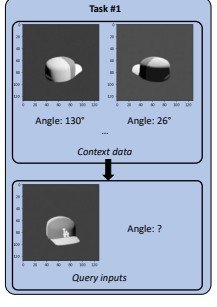
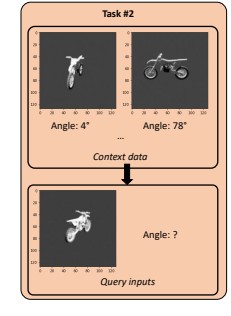
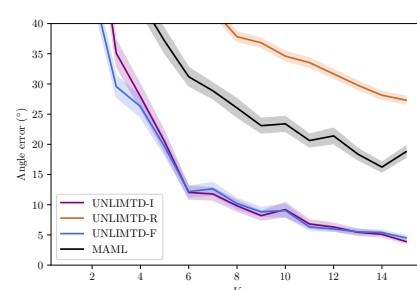

Figure 9: Left: Examples of vision tasks. Right: angle error as a function of the number of context datapoints $K$, after different meta-trainings on Shapenet1D (UNLIMITD-I, UNLIMITD-R and UNLIMITD-F vs MAML). Note how UNLIMITD-I and UNLIMITD-F make better predictions than MAML. Also note that contrary to the small network case, UNLIMITD-R performs worse than MAML and than the two other variants.

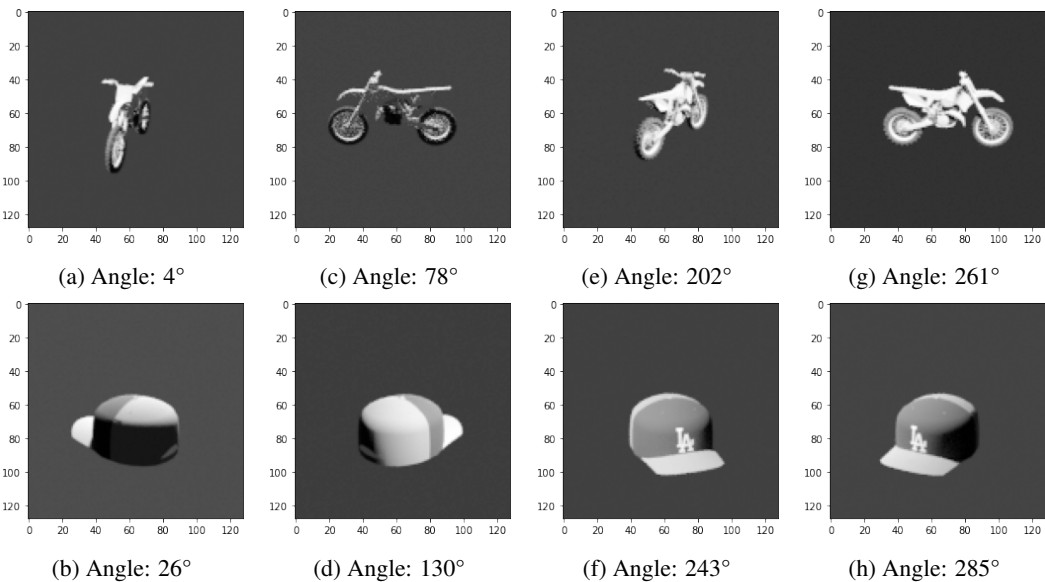

Figure 10: Examples of images and ground-truth angles from the Shapenet1D dataset (Gao et al., 2022)

