# OpenReview forum: "Uncertainty-Aware Meta-Learning for Multimodal Task Distributions"
_ICLR.cc/2023/Conference — Submitted to ICLR 2023_

### Official Review · Reviewer_J2Ni · 2022-10-17

**Confidence:** 4
**Correctness:** 4
**Technical Novelty And Significance:** 3
**Empirical Novelty And Significance:** 2
**Recommendation:** 5

**Clarity, Quality, Novelty And Reproducibility:**

The paper is well written and easy to follow. Within its scope, most discussion are detailed and well-motivated. Some aspects of the proposed method are novel. Source code is provided for reproducibility.

**Strength And Weaknesses:**

Strength:
1. The paper is well-written and easy to follow. All elements of the method are well-motivated. The GP-based approach is sound.
2. Detailed discussion is presented for the practical algorithm, highlighting the trade-offs and design constraints.
3. The empirical comparison with MAML is detailed and shows improved performance.

Weakness:
1. The method so far only tackles regression problem, excluding a large segment of the meta-learning literature on classification. In principle, it should be straightforward to transform classification into regression (see the ridge regression approach in [1, 2]), thus allowing the proposed method to be compared to wider sets of baselines in more settings.
2. Related to 1, the empirical comparison is very limited. Only MAML is used as the baseline. It is also unclear why OoD detection is only evaluated among the different variants of the proposed method, rather against baselines discussed in related work section.
3. Computational requirements should be better discussed with respect to other methods. For instance, does the method require second-order derivatives during optimization? If so, does this either 1) limit the size of the network or 2) require multiple-gpus or TPU to optimize? The network used for the experiments should be mentioned in the paper.

[1] The Role of Global Labels in Few-Shot Classification and How to Infer Them, Wang et. al, Neurips 2021

[2] Meta-learning with differentiable closed-form solvers, Bertinetto et. al, ICLR 20119




**Summary Of The Paper:**

The paper proposes a Gaussian Process (GP)-based approach to meta-learning. Using GP, the method allows probabilistic predictions for in-distribution tasks and detection for out-of-distribution (OoD) tasks. Several practical algorithm variants are introduced to tackle computational requirements of GP. Empirically, the proposed method is compared to MAML on regression problems and show improved performance.

**Summary Of The Review:**

The proposed method introduces a GP-based approach to meta-learning, which grants several desirable properties including probabilistic prediction and OoD task detection. The approach appears well-motivated and sound. However, current empirical evaluation is limited: only MAML baseline is considered and only regression setting is considered. It is difficult to assess the efficacy of the proposed method compared to the broader literature.

---

### Official Review · Reviewer_pWMB · 2022-10-23

**Confidence:** 4
**Correctness:** 3
**Technical Novelty And Significance:** 3
**Empirical Novelty And Significance:** 2
**Recommendation:** 3

**Clarity, Quality, Novelty And Reproducibility:**

The paper is well written and clear. The work is original. Code is provided to reproduce the results, additional details on the experiments are reported in the appendix.

**Details Of Ethics Concerns:**

None.

**Strength And Weaknesses:**



Strength
--------

- The method can be used in a variety of settings, which makes it flexible.

- The possibility of quantifying uncertainty is a plus, this makes the method superior to classical meta-learning approaches like MAML.

- The method is able to deal with the challenging multimodal task distribution setting, which differentiates it from previous probabilistic methods.


Weaknesses
----------

1) My main concerns regard the empirical evaluation, in particular the tasks used. The sine-wave experiments are a basic way to evaluate the performance of the proposed method. A more challenging version of the sine-wave is the out-of-range condition, where some methods have failed to generalize, for more details see Patacchiola et al. (2020). Overall, the empirical evaluation needs to be bulked up. Reporting the results on a larger set of regression problems would be beneficial, e.g. see the few-shot datasets used in Sendera et al. (2021).

2) Another concern regards the baselines used in the experiments. The authors compare against MAML (e.g. see Figure 3), which is a pretty poor baseline for the regression case. The authors disregarded a line of work rooted in probabilistic and Bayesian theory that has been quite successful in dealing with unimodal task distributions in meta-learning. Those methods are neither mentioned nor discussed in the paper. The only exception is a good comparison with ALPaCA (Harrison et al. 2018) provided in the appendix. Notable examples of paper that should be discussed and compared against are R2-D2 (Bertinetto et al. 2018), ADKL (Tossou et al. 2019), Deep Kernel Transfer (Patacchiola et al. 2020), and NGGPs (Sendera et al. 2021). In particular, Patacchiola et al. (2020) provide a table that compares all these methods on the sine-wave experiments (in-range and out-of-range conditions). It is important to see how the proposed method compares against these stronger baselines.

3) Working in weight space can be problematic, in particular when representing the covariance matrix. This has been addressed by the authors in Section 4.1.1. My concern here is that using a low-dimensional representation of the covariance may be appropriate in some cases but it may not be expressive enough in others. Given the limited set of experiments and backbones tested, it is hard to get a grasp on this point. For instance, it is not clear how the expressiveness of the covariance-matrix is affected by using neural networks of different size/type. I could not find a satisfying report on this, just a few lines in Section 6 (paragraph "Unimodal meta-learning"). It would be useful to see how the performance of the I-R-F variants change when using models of different size (while keeping fixed the dataset) and with deeper backbones (e.g. in visual regression problems).

4) A corollary of the previous point is the computational complexity of the method, in terms of both time and space. In particular, it would be useful to see how the computational complexity of the I-R-F variants changes when using common backbones and how it compares to other probabilistic methods. This is quite important here, as it can be a major bottleneck that hinder scalability.

5) Minor corrections. In Equation 4 there are unmatched brackets. Figure 3 is hard to read, especially when it comes to the values on the right side where the curves are very close, a tabular version may be better. A brief description of the variants of UNLIMITD called I-R-F should be added earlier in the manuscript to improve clarity (e.g. at the end of the Introduction).


References
----------

Bertinetto, L., Henriques, J. F., Torr, P. H., & Vedaldi, A. (2018). Meta-learning with differentiable closed-form solvers. arXiv preprint arXiv:1805.08136.

Harrison, J., Sharma, A., & Pavone, M. (2018, December). Meta-learning priors for efficient online bayesian regression. In International Workshop on the Algorithmic Foundations of Robotics (pp. 318-337). Springer, Cham.

Patacchiola, M., Turner, J., Crowley, E. J., O'Boyle, M., & Storkey, A. J. (2020). Bayesian meta-learning for the few-shot setting via deep kernels. Advances in Neural Information Processing Systems, 33, 16108-16118.

Sendera, M., Tabor, J., Nowak, A., Bedychaj, A., Patacchiola, M., Trzcinski, T., ... & Zieba, M. (2021). Non-Gaussian Gaussian Processes for Few-Shot Regression. Advances in Neural Information Processing Systems, 34, 10285-10298.

Tossou, P., Dura, B., Laviolette, F., Marchand, M., & Lacoste, A. (2019). Adaptive deep kernel learning. arXiv preprint arXiv:1905.12131.


**Summary Of The Paper:**

This paper proposes a method  that is able to deal with regression problems in a meta-learning framework. The method estimates a parametric and tuneable distribution, leveraging Bayesian inference with linearized neural networks. The method is flexible and able to deal with unimodal and multimodal task distributions. Experiments on sine-waves prediction are provided to showcase the effectiveness of the proposed solution.

**Summary Of The Review:**

The paper provides an interesting angle on the few-shot regression problem, which is significantly different from previous work. However, the limited empirical evaluation, shortage of baselines, and possible scalability issues, arise serious concerns that need to be addressed in the rebuttal.

---

### Official Review · Reviewer_xBkp · 2022-10-25

**Confidence:** 3
**Correctness:** 3
**Technical Novelty And Significance:** 3
**Empirical Novelty And Significance:** 3
**Recommendation:** 6

**Clarity, Quality, Novelty And Reproducibility:**

The paper is well-written. Even though the idea is not very novel, but the proposed method makes sense.

**Strength And Weaknesses:**

The main strengths:
1.It models the true distribution of tasks by performing Bayesian inference on a linearized neural network.
2.By defining different covariances in the prior distribution, three variants are proposed:  identity covariance, low-dimensional covariance with random directions, and covariance based on Fisher information matrix, which can balance scalability and expressivity of the model.
3.The experimental results perform well. The MSE is small which shows the accurate probability prediction on in-distribution tasks. The AUC-ROC scores also demonstrate the effective detection of context data from OoD tasks.

The main weaknesses:
1.The review of related work on uncertainty in meta-learning is small and it is difficult to locate the main contribution of this paper.
2.The Ood detection only extends from classification to regression, which is not very innovative.
3.The experiments only focus on the comparison between various models proposed by authors and lack of comparison with other methods.

**Summary Of The Paper:**

This paper presents an uncertainty-aware meta-learning method for multi-modal task distributions.
It utilizes a learnable parametric distribution on a linearized neural network to model the real distribution over tasks. The method uses Bayesian inference to achieve uncertainty-aware adaptivity and learns multi-modal task distributions through a mixture of Gaussian processes.

What contributions does it make:
1.It combines meta-learning and uncertainty from a probabilistic perspective.
2.It makes efficient probabilistic prediction on in-distribution tasks by parametrically learning prior distributions, while efficiently detecting OoD context data at test time.
3.It learns multi-modal task distributions through mixed Gaussian processes.

**Summary Of The Review:**

The method uses Bayesian inference to achieve uncertainty-aware adaptivity and learns multi-modal task distributions through a mixture of Gaussian processes. The idea makes sense and the experimental results perform well.

---

### Official Review · Reviewer_eEkt · 2022-10-29

**Confidence:** 2
**Correctness:** 3
**Technical Novelty And Significance:** 2
**Empirical Novelty And Significance:** 2
**Recommendation:** 3

**Clarity, Quality, Novelty And Reproducibility:**

Clarity and quality is good, but novelty and significance is limited as the authors did not compare against the existing baselines.

**Strength And Weaknesses:**

Strength
- Paper is well written
- Method has been developed well based on theoretically solid Bayesian regression and Gaussian processes.

Weaknesses
- The experimental results are all toy examples, such as sinusoidal regression and mixture of simple regression problems. Those experimental settings are far from real-world scenarios and also the amount itself is also very limited.
- Insufficient baselins. There are many works that solves exactly the same problem, including all the Bayesian version of MAML and probabilistic versions of Prototypical-like networks. But they only compare agains themselves or simple vanilla MAML baselines. To name a few, see the below references. There should be tons of more methods the authors should compare against.

References
- Harrison et al., Meta-Learning Priors for Efficient Online Bayesian Regression
- Kim et al., Bayesian Model-Agnostic Meta-Learning
- Finn et al., Probabilistic Model-Agnostic Meta-Learning
- Gordon et al., Meta-Learning Probabilistic Inference For Prediction
- Willette et al., Meta-Learning Low Rank Covariance Factors for Energy-Based Deterministic Uncertainty



**Summary Of The Paper:**

This paper propose to tackle limited data scenario in meta-learning setting. They make use of Bayesian linear regression and Gaussian processes to model uncertainty when the data is limited. The experimental results demonstrate the efficacy of their method.

**Summary Of The Review:**

In summary, the authors proposed interesting methods for meta-learning for limited data regime, but the experimental results are very limited. Therefore, I recommend reject.

---

### Decision · Program_Chairs · 2023-01-20

**Decision:**

Reject

**Justification For Why Not Higher Score:**

N/A

**Justification For Why Not Lower Score:**

N/A

**Metareview: Summary, Strengths And Weaknesses:**

This paper proposed a new meta-learning algorithm that makes probabilistic predictions efficiently, detects out-of-distribution context data, and performs well on heterogeneous, multimodal task distributions. While the paper contains some interesting idea, reviewers raised major weakness concerns about weak empirical evaluations, lack of convincing experimental results and strong baselines. Overall, the quality of this work is below the acceptance bar and the authors did not give any response to address the review questions.